# Identification and Estimation of the Bi-Directional MR with Some Invalid Instruments

**Feng Xie[1], Zhen Yao[1], Lin Xie[1], Yan Zeng[1],*, Zhi Geng[1]**

[1]Beijing Technology and Business University

## Abstract

We consider the challenging problem of estimating causal effects from purely observational data in the bi-directional Mendelian randomization (MR), where some invalid instruments, as well as unmeasured confounding, usually exist. To address this problem, most existing methods attempt to find proper valid instrumental variables (IVs) for the target causal effect by expert knowledge or by assuming that the causal model is a one-directional MR model. As such, in this paper, we first theoretically investigate the identification of the bi-directional MR from observational data. In particular, we provide necessary and sufficient conditions under which valid IV sets are correctly identified such that the bi-directional MR model is identifiable, including the causal directions of a pair of phenotypes (i.e., the treatment and outcome). Moreover, based on the identification theory, we develop a cluster fusion-like method to discover valid IV sets and estimate the causal effects of interest. We theoretically demonstrate the correctness of the proposed algorithm. Experimental results show the effectiveness of our method for estimating causal effects in both one-directional and bi-directional MR models.

## 1 Introduction

Mendelian randomization (MR) is a powerful method for estimating the causal effect of a potentially modifiable risk factor (treatment) on disease (outcome) in observational studies, which has been used across a wide variety of contexts, such as clinic [Mokry et al., 2015, Carreras-Torres et al., 2017], socioeconomics [Cuellar-Partida et al., 2016, Brumpton et al., 2020], and drug targets [Ference et al., 2015, Li et al., 2020]. Most existing MR methods assume a one-directional causal relationship. However, bi-directional relationships are ubiquitous in real-life scenarios, for example, obesity and vitamin D status [Vimaleswaran et al., 2013], body mass index (BMI) and type 2 diabetes (T2D), etc. Understanding and analyzing bi-directional relationships can provide deeper insights into complex systems, leading to more effective interventions and solutions.

Within a causal inference framework, MR can be implemented as a form of instrumental variables (IVs) analysis in which genetic variants[2] serve as IVs for the modifiable risk factor. Specifically, as shown in Figure 1, suppose obesity ($X$) and Vitamin D status ($Y$) are the treatment and outcome of interest, respectively, and lifestyle factors ($\mathbf{U}$) are the unmeasured confounders between them. One can use genetic variants $\mathbf{G}$ as IVs to explore the causal effect of $X$ on $Y$ in observational data [Vimaleswaran et al., 2013] if $\mathbf{G}$ are valid IVs, i.e., $\mathbf{G}$ satisfy the following three assumptions [Wright, 1928, Goldberger, 1972, Bowden and Turkington, 1990, Hernán and Robins, 2006]:

**A1.** [*Relevance*] The genetic variants $\mathbf{G}$ are associated with the exposure $X$;

**A2.** [*Exclusion Restriction*] The genetic variants $\mathbf{G}$ have no direct pathway to the outcome $Y$;

---

*Corresponding author

[2]The genetic variants that act as IVs in MR studies are Single Nucleotide Polymorphisms (SNPs).

38th Conference on Neural Information Processing Systems (NeurIPS 2024).

**A3.** [*Randomness*] The genetic variants **G** are uncorrelated with unmeasured confounders **U**.

Figure 1 illustrates these three assumptions for a genetic variant $G$ that is a valid IV relative to the causal relationship $X \rightarrow Y$. Given a valid IV $G$ in the linear model, one can estimate the causal effect of risk factor $X$ on the outcome $Y$ of interest consistently by using Two-Stage Least Squares (TSLS) [Burgess et al., 2017]. However, in most MR studies, the genetic variants may include some invalid IVs due to horizontal pleiotropy− where some genetic variants may have pleiotropic effects on both the treatment $X$ and the outcome $Y$ via different biological pathways [Davey Smith and Ebrahim, 2003, Lawlor et al., 2008]. How to select valid IVs from such genetic variants becomes the major issue in the MR study. In general, valid IVs need to be chosen based on

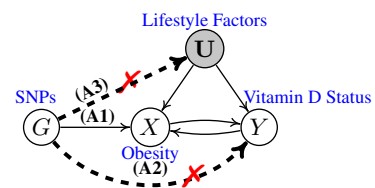

Figure 1: Graphical illustration of a valid IV model, where dashed lines indicate the absence of arrows. $G$ is a valid IV relative to the causal relationship $X \rightarrow Y$.

domain knowledge or expert experience [Richmond et al., 2017, Martens et al., 2006, Zhao et al., 2023]. However, it is usually difficult to select appropriate valid IVs without precise prior knowledge of genetic variants, and an invalid IV may cause biased estimation of the effect of $X$ on $Y$ [Sanderson et al., 2022, Richmond and Smith, 2022]. Therefore, it is desirable to investigate ways of selecting genetic variants as valid IVs solely from observed variables.

Many contributions have been made to handle MR studies with some invalid IVs under specific assumptions. Existing methods can be roughly divided into four strategies. The first strategy assumes the *InSIDE assumption*, which states that the genetic variants' pleiotropic effects on the outcome $Y$ are uncorrelated with their effects on the exposure $X$, and requires that the number of IVs increases with the sample size [Bowden et al., 2015]. The second strategy is based on the *Majority rule assumption*, which assumes that over $50\%$ of the candidate IVs are valid [Kang et al., 2016, Bowden et al., 2016, Windmeijer et al., 2019, Hartford et al., 2021]. The third strategy assumes the *Plurality rule assumption*, which means that the number of valid IVs is larger than any number of invalid IVs with the same ratio estimator limit [Guo et al., 2018, Windmeijer et al., 2021]. The last strategy assumes the *Two-Valid IVs assumption*, which requires that the number of valid IVs is greater than or equal to 2, and the *Rank-faithfulness assumption*, which states that the probability distribution $P$ is rank-faithful to a directed acyclic graph $\mathcal{G}$ if every rank constraint on a sub-covariance matrix that holds in $P$ is entailed by every linear structural model with respect to $\mathcal{G}$ [Silva and Shimizu, 2017, Xie et al., 2022, Cheng et al., 2023]. Although these methods have been used in various fields, they assumed that the one-directional causal relationship between $X$ and $Y$ is known and may fail to handle scenarios where bi-directional relationships are present.

Recently, Darrous et al. [2021] proposed a latent heritable confounder MR method to estimate bi-directional causal effects with a hierarchical prior on the true parameters. However, their method is sensitive to prior misspecification. Later, Li and Ye [2022] introduced a focusing framework to test bi-directional causal effects under the *noise distribution of the effect estimates assumption*, which is exciting. However, this assumption is strictly parameterized (See Condition 1 in [Li and Ye, 2022] for more details). Moreover, they focus on the two-sample MR model that assumes homogeneity across samples, whereas we consider the one-sample MR model. See Appendix A for more details.

In this paper, we aim to investigate the identifiability of the one-sample bi-directional MR model and to provide an estimation method. Specifically, we make the following contributions:

1. We present sufficient and necessary conditions for the identifiability of the bi-directional MR model, enabling both valid IV sets for each direction and the causal effects of interests to be correctly identified.
2. We propose a practical and effective cluster fusion-like algorithm for unbiased estimation. This algorithm correctly identifies which IVs are valid, determines the causal directions to which the valid IV sets belong, and estimates the bi-directional causal effects.
3. Extensive experiments demonstrate the efficacy of the proposed algorithms in MR models.

## 2 Model Definition and IV Estimator

### 2.1 Bi-Directional MR Causal Models

Suppose we have $n$ independent samples $(X_i, Y_i, \mathbf{G}_i)$ from the distribution of $(X, Y, \mathbf{G})$, where $X \in \mathbb{R}$ and $Y \in \mathbb{R}$ is a pair of phenotypes of interest, and $\mathbf{G} = (G_1, G_2, \ldots, G_g)^\intercal \in \mathbb{R}^g$ denotes

measured genetic variants, which may include invalid IVs. Analogous to Li and Ye [2022], for each sample, indexed by $i$, the phenotype $X_i$ is taken as a linear function of the phenotype $Y_i$, genetic variants $\mathbf{G}_i$, and an error term $\varepsilon_{X_i}$, while phenotype $Y_i$ is modeled as a linear function of the phenotype $X_i$, genetic variants $\mathbf{G}_i$, and an error term $\varepsilon_{Y_i}$. Without loss of generality, we assume that all variables have a zero mean (otherwise can be centered). Specifically, the generating process of the data is given below:

$$
\begin{aligned}
X &= Y\beta_{Y \to X} + \mathbf{G}^\mathsf{T}\boldsymbol{\gamma}_X + \varepsilon_X, \\
Y &= X\beta_{X \to Y} + \mathbf{G}^\mathsf{T}\boldsymbol{\gamma}_Y + \varepsilon_Y,
\end{aligned}
\tag{1}
$$

where $\beta_{Y \to X}$ is the causal effect of $Y$ on $X$ and $\beta_{X \to Y}$ is that of $X$ on $Y$. $\boldsymbol{\gamma}_X = (\gamma_{X,1}, ..., \gamma_{X,g})^\mathsf{T}$ and $\boldsymbol{\gamma}_Y = (\gamma_{Y,1}, ..., \gamma_{Y,g})^\mathsf{T}$ are the direct effects of genetic variants $\mathbf{G}$ on $X$ and $Y$, respectively. Note that $\varepsilon_X$ and $\varepsilon_Y$ are error terms that are correlated due to unmeasured confounders $\mathbf{U}$ between $X$ and $Y$. Besides, there may exist a single genetic variant $G_i \in \mathbf{G}$ with both $\gamma_{X,i} \neq 0$ and $\gamma_{Y,i} \neq 0$, which implies that $G_i$ also has a direct pathway to the outcome, violating Assumption A2 [*Exclusion Restriction*]. Like Li and Ye [2022], we here restrict the independence between any two genetic variants in $\mathbf{G}$ (i.e., genetic variants are randomized). In Section 5, we discuss the situation where this restriction is violated. Unlike Li and Ye [2022], we allow genetic variants to cause unmeasured confounders (partially violating Assumption A3 [*Randomness*]).

Following Hausman [1983], we assume that $\beta_{X \to Y}\beta_{Y \to X} \neq 1$. Then let $\Delta = 1/(1 - \beta_{X \to Y}\beta_{Y \to X})$, Eq.(1) can be reorganized to avoid recursive formulations as follows:

$$
\begin{aligned}
X &= (\mathbf{G}^\mathsf{T}\boldsymbol{\gamma}_X + \mathbf{G}^\mathsf{T}\boldsymbol{\gamma}_Y\beta_{Y \to X} + \varepsilon_X + \varepsilon_Y\beta_{Y \to X})\Delta, \\
Y &= (\mathbf{G}^\mathsf{T}\boldsymbol{\gamma}_X\beta_{X \to Y} + \mathbf{G}^\mathsf{T}\boldsymbol{\gamma}_Y + \varepsilon_X\beta_{X \to Y} + \varepsilon_Y)\Delta.
\end{aligned}
\tag{2}
$$

If $\beta_{X \to Y}\beta_{Y \to X} = 1$, the causal effects $\beta_{X \to Y}$ and $\beta_{Y \to X}$ are not identifiable, even in the presence of a valid IV.

*Goal.* In this paper, we aim to address whether the bi-directional MR model in Eq.(1) is identifiable; that is, whether we can estimate the causal effects $\beta_{X \to Y}$ and $\beta_{Y \to X}$ given infinite data, even without prior knowledge about which candidate IVs are valid or invalid.

## 2.2 IV Estimator

In this section, we briefly describe the classical IV estimator, the two-stage least squares (TSLS) estimator, which is capable of consistently estimating the causal effects $\beta_{X \to Y}$ and $\beta_{Y \to X}$ of interest in the above bi-directional MR causal model of Eq.(1) when the valid IVs are known [Wooldridge, 2010]. We denote by $\mathbf{G}_{\mathcal{V}}^{X \to Y}$ the set consisting of all valid IVs in $\mathbf{G}$ relative to $X \to Y$, and $\mathbf{G}_{\mathcal{I}}^{X \to Y}$ the set that consists of at least one invalid IV in $\mathbf{G}$ relative to $X \to Y$. Similar notations are applied to the inverse causal relationship $Y \to X$, i.e., $\mathbf{G}_{\mathcal{V}}^{Y \to X}$ and $\mathbf{G}_{\mathcal{I}}^{Y \to X}$.

**Proposition 1** (Two Stage Least Square (TSLS) Estimator)**.** *Assume the system is a linear bi-directional causal model 1. For a given causal relationship $X \to Y$ in the system, the causal effect of $X$ on $Y$ can be identified by*

$$
\hat{\beta}_{X \to Y} = [X^\mathsf{T}\mathbf{P}X]^{-1} X^\mathsf{T}\mathbf{P}Y = \beta_{X \to Y},
\tag{3}
$$

*where $\mathbf{P} = (\mathbf{G}_{\mathcal{V}}^{X \to Y})^\mathsf{T} \left[ \mathbf{G}_{\mathcal{V}}^{X \to Y}(\mathbf{G}_{\mathcal{V}}^{X \to Y})^\mathsf{T} \right]^{-1} \mathbf{G}_{\mathcal{V}}^{X \to Y}$ is the projection matrix.*

**Remark 1.** *Note that, when utilizing the TSLS estimator, the causal effect is biased if the candidate set $\mathbf{G}$ includes invalid IVs. For instance, consider the causal relationship $X \to Y$, given an invalid IV set $\mathbf{G}_{\mathcal{I}}^{X \to Y}$, the causal effect of $X$ on $Y$ is given by*

$$
\hat{\beta}_{X \to Y} = \left[ X^\mathsf{T}\tilde{\mathbf{P}}X \right]^{-1} X^\mathsf{T}\tilde{\mathbf{P}}Y \quad = \beta_{X \to Y} + \underbrace{\left[ X^\mathsf{T}\tilde{\mathbf{P}}X \right]^{-1} X^\mathsf{T}\tilde{\mathbf{P}}(\mathbf{G}^\mathsf{T}\boldsymbol{\gamma}_Y + \varepsilon_Y)}_{\beta_{bias}},
$$

*where $\tilde{\mathbf{P}} = (\mathbf{G}_{\mathcal{I}}^{X \to Y})^\mathsf{T} \left[ \mathbf{G}_{\mathcal{I}}^{X \to Y}(\mathbf{G}_{\mathcal{I}}^{X \to Y})^\mathsf{T} \right]^{-1} \mathbf{G}_{\mathcal{I}}^{X \to Y}$.*

In the remaining of the paper, we denote such a procedure by $\hat{\beta}_{X \to Y} = \text{TSLS}(X, Y, \mathbf{G}^{X \to Y})$. Similarly, for causal relationship $Y \to X$, one can also have the corresponding causal effect of $Y$ on $X$ given the set of IVs, i.e., $\hat{\beta}_{Y \to X} = \text{TSLS}(Y, X, \mathbf{G}^{Y \to X})$. For details about the proof of Proposition 1 and Remark 1, please refer to Appendix G.1 and G.2.

# 3 Identifiability of Bi-Directional MR Model

In this section, we first show by a simple example that the valid and invalid IV sets can impose different constraints. Then, we formulate these constraints and present the necessary and sufficient conditions that render the model identifiable.

## 3.1 A Motivating Example

Figure 2 gives an illustration of our basic idea. Suppose there are 2 valid IVs, i.e., $G_1$ and $G_3$, and 3 invalid IVs, i.e., $G_2$, $G_4$ and $G_5$, relative to the causal relationship $X \to Y$. For the inverse $Y \to X$, identical conclusions can be derived. Assume the generating process of $X$ and $Y$ follows Eq.(1), where specifically we set all variables to have a zero mean and unit variance and $X = Y\beta_{Y \to X} + G_1\gamma_{X,1} + G_2\gamma_{X,2} + G_3\gamma_{X,3} + G_4\gamma_{X,4} + G_5\gamma_{X,5} + \varepsilon_X$, and $Y = X\beta_{X \to Y} + G_2\gamma_{Y,2} + G_4\gamma_{Y,4} + G_5\gamma_{Y,5} + \varepsilon_Y$, where $\beta_{X \to Y} = \beta_{Y \to X} = 0.6, \gamma_{X,1} = \gamma_{X,2} = \gamma_{X,3} = 1, \gamma_{X,4} = 1.8, \gamma_{X,5} = 1.2, \gamma_{Y,2} = 0.5, \gamma_{Y,4} = 0.6, \gamma_{Y,5} = 0.3$.

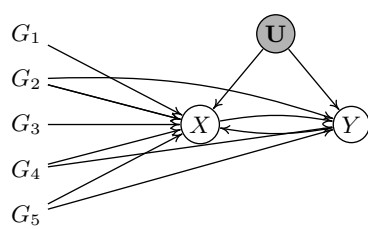

Figure 2: An illustrative example where valid and invalid IV sets induce distinct constraints, where $\mathbf{G}_{\mathcal{V}}^{X \to Y} = (G_1, G_3)^{\mathsf{T}}$ is a valid IV set, while $\mathbf{G}_{\mathcal{I}}^{X \to Y} = (G_2, G_4, G_5)^{\mathsf{T}}$ is invalid due to pathways $G_2 \to Y$, $G_4 \to Y$ and $G_5 \to Y$.

Interestingly, we have the following observations:

**Observation 1.** Given a valid IV set where there are at least two IVs , e.g., $\mathbf{G}_{\mathcal{V}}^{X \to Y} = (G_1, G_3)^{\mathsf{T}}$, we have:

$$\text{corr}(Y - X\omega_{\{G_3\}}, G_1) = 0, \qquad \text{corr}(Y - X\omega_{\{G_1\}}, G_3) = 0,$$

where $\text{corr}(\cdot)$ denotes the Pearson's correlation coefficient between two random variables, and $\omega_{\{G_i\}} = \text{TSLS}(X, Y, \{G_i\})$ with $i \in \{1, 3\}$.

**Observation 2.** However, given an invalid IV set, e,g., $\mathbf{G}_{\mathcal{I}}^{X \to Y} = (G_2, G_4, G_5)^{\mathsf{T}}$, we have:

$$\text{corr}(Y - X\omega_{\{G_4,G_5\}}, G_2) \neq 0, \text{corr}(Y - X\omega_{\{G_2,G_5\}}, G_4) \neq 0, \text{corr}(Y - X\omega_{\{G_2,G_4\}}, G_5) \neq 0,$$

where $\omega_{\{G_i,G_j\}} = \text{TSLS}(X, Y, \{G_i, G_j\})$ with $i \neq j$ and $i, j \in \{2, 4, 5\}$.

Based on these two observations, we can see that the valid and invalid IV sets entail clearly different constraints regarding the correlation coefficients[3], which inspires us to further derive our identification analysis. For details about the observations, please refer to Appendix C.

## 3.2 Main Identification Results

We begin with the following definition and the main assumptions.

**Definition 1.** *(Pseudo-Residual) Let $\mathbb{G}$ be a subset of candidate genetic variants. A pseudo-residual of $\{X, Y\}$ relative to $\mathbb{G}$ is defined as*

$$\mathcal{PR}_{(X,Y \mid \mathbb{G})} := Y - X\omega_{\mathbb{G}}, \tag{4}$$

*where $\omega_{\mathbb{G}} = \text{TSLS}(X, Y, \mathbb{G})$.*

This "pseudo-residual" is applied to measure the uncorrelated relationship with some genetic variants. Note that concepts to "pseudo-residual" have been developed to address different tasks [Drton and Richardson, 2004, Chen et al., 2017, Cai et al., 2019, Xie et al., 2020], but our formalization is different from theirs in terms of the parameter $\omega_{\mathbb{G}}$. To the best of our knowledge, it has not been realized that the uncorrelated property involving such pseudo-residuals reflects the validity of the IVs in the bi-directional MR causal model (see Propositions $2 \sim 3$).

As analyzed by Chu et al. [2001], within the framework of linear models, a variable being a valid IV imposes no constraints on the joint marginal distribution of the observed variables. In other words, there is no available test to determine whether a variable is a valid IV without making further assumptions. Therefore, we introduce the following assumption.

**Assumption 1** (Valid IV Set). *For a given causal relationship (if the relationship exists), there exists a valid IV set that consists of at least two valid IVs. For example, for the causal relationship $X \to Y$, $|\mathbf{G}_{\mathcal{V}}^{X \to Y}| \geq 2$.*

---

[3]Please note that these constraints can be tested easily by observed data.

Assumption 1 is the same as in [Silva and Shimizu, 2017, Cheng et al., 2023] if the model is a one-directional MR model. Notice that this assumption is much milder than the *Majority rule* assumption [Kang et al., 2016, Bowden et al., 2016, Windmeijer et al., 2019, Hartford et al., 2021] and *Plurality rule* assumption [Guo et al., 2018, Windmeijer et al., 2021]: Assumption 1 does not rely on the total number of genetic variants, while *Majority rule* and *Plurality rule* assumptions do.

We now show that under Assumption 1, one can identify the class of invalid IV sets.

**Proposition 2** (Identifying Invalid IV Sets). *Let $\mathbb{G} = \{G_1, \ldots, G_p\}, p \geq 2$ be a subset of candidate genetic variants **G**. Suppose that Assumption 1 and the size of participants $n \to \infty$ hold. If there exists a $G_j \in \mathbb{G}$ such that,*

$$\mathrm{corr}(\mathcal{PR}_{(X,Y\,|\,\mathbb{G}\setminus G_j)}, G_j) \neq 0, \tag{5}$$

*then $\mathbb{G}$ is an invalid IV set for any one of the causal relationships.*

Proposition 2 is valuable when given a set of genetic variants but lack prior knowledge of the validity of IV sets. By testing specific correlations in Eq.(5), we can conclude that the current IV set is invalid.

**Example 1.** *Consider the example in Figure 2. Suppose $\mathbb{G} = \{G_1, G_2\}$, according to Proposition 2 we have $\mathrm{corr}(\mathcal{PR}_{(X,Y\,|\,\{G_2\})}, G_1) \neq 0$. Thus $\mathbb{G}$ is an invalid IV set.*

One may naturally raise the following question: Can all the invalid IV sets be identified by Proposition 2? In other words, can we identify all the valid IV sets? Unfortunately, Assumption 1 is an insufficient condition for identifying the sets of valid IVs. See an illustrative example below.

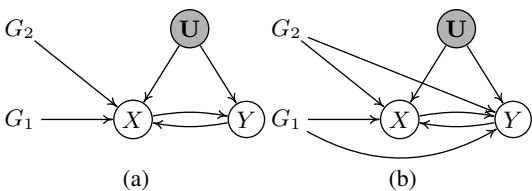

(a)          (b)

**Example 2** (Counterexample). *Consider the example in Figure 3. Suppose $\mathbb{G} = \{G_1, G_2\}$. In subgraph (a), for $G_1 \in \mathbb{G}$, according to Proposition 2 we obtain*

Figure 3: An illustrative example that valid and invalid IV sets may induce the same constraints in Eq.(5) of Proposition 2, where $\{G_1, G_2\}$ is the set of valid IVs in (a), while invalid in (b). It implies Assumption 1 is not sufficient to find valid IV sets.

$$\mathrm{corr}(\mathcal{PR}_{(X,Y\,|\,\{G_2\})}, G_1) = 0 \tag{6}$$

*Similarly, for $G_2 \in \mathbb{G}$, we have*

$$\mathrm{corr}(\mathcal{PR}_{(X,Y\,|\,\{G_1\})}, G_2) = 0 \tag{7}$$

*In subgraph (b), for $G_1 \in \mathbb{G}$, we check the condition of Proposition 2 and obtain*

$$\mathrm{corr}(\mathcal{PR}_{(X,Y\,|\,\{G_2\})}, G_1) = \frac{\gamma_{Y,1}\gamma_{X,2} - \gamma_{Y,2}\gamma_{X,1}}{\beta_{Y \to X}\gamma_{Y,2} + \gamma_{X,2}}. \tag{8}$$

*Similarly, for $G_2 \in \mathbb{G}$, we have*

$$\mathrm{corr}(\mathcal{PR}_{(X,Y\,|\,\{G_1\})}, G_2) = \frac{\gamma_{Y,2}\gamma_{X,1} - \gamma_{Y,1}\gamma_{X,2}}{\beta_{Y \to X}\gamma_{Y,1} + \gamma_{X,1}}. \tag{9}$$

*We know that $\{G_1, G_2\}$ is an invalid IV set in this case. However, if the proportion of effects of $G_1$ on $X$ and $Y$ is equal to the proportion of the effects of $G_2$ on $X$ and $Y$, i.e., $\gamma_{X,1}\gamma_{Y,2} = \gamma_{Y,1}\gamma_{X,2}$, we have*

$$\mathrm{corr}(\mathcal{PR}_{(X,Y\,|\,\{G_2\})}, G_1) = 0, \qquad \mathrm{corr}(\mathcal{PR}_{(X,Y\,|\,\{G_1\})}, G_2) = 0,$$

*which is consistent with the conclusion of subgraph (a). These observations imply that one can not identify valid IV sets under only Assumption 1.*

To give sufficient conditions for identifying valid IV sets, we first introduce the following assumption.

**Assumption 2** (Generic Identifiability). *For a given MR causal model, parameters $\gamma$ and $\beta$ live in a set of Lebesgue measure non-zero.*

In other words, Assumption 2 specifically implies that given a general subset of candidate genetic variants $\mathbb{G}$, there exists a $G_j \in \mathbb{G}$ that satisfies $|\sum_{G_i \in \mathbb{G}/G_j} \bar{\beta}_i cov(G_i, G_i)[(\gamma_{X,i} + \gamma_{X,U}\gamma_{U,i})(\gamma_{Y,j} + \gamma_{Y,U}\gamma_{U,j}) - (\gamma_{Y,i} + \gamma_{Y,U}\gamma_{U,i})(\gamma_{X,j} + \gamma_{X,U}\gamma_{U,j})]| \neq 0$, where $\bar{\beta}_i \neq 0$ is the estimated coefficient between $G_i$ and $X$. For instance, in Example 2, such an assumption simplifies to $\gamma_{X,1}\gamma_{Y,2} \neq \gamma_{Y,1}\gamma_{X,2}$. Whenever $\gamma_{X,1}\gamma_{Y,2} = \gamma_{Y,1}\gamma_{X,2}$, parameters $\gamma$ and $\beta$ would fall into a set of Lebesgue measure zero, rendering non-identifiable of the model. It is worth noting that in real life, although we cannot obtain the ground truths of causal effects between variables, especially when latent confounders are involved, Assumption 2 is generally satisfied. This is because the set of conditions to violate it occupies a very small portion of the entire space.

With Assumption 2, we now derive sufficient conditions for generic identifiability in Proposition 3.

**Proposition 3** (Identifying Valid IV Sets). *Let $\mathbb{G} = \{G_1, \ldots, G_p\}, p \geq 2$ be a subset of candidate genetic variants* **G**. *Suppose that Assumptions 1 and 2, and the size of participants $n \to \infty$ hold. If each $G_j \in \mathbb{G}$ satisfies*

$$\text{corr}(\mathcal{PR}_{(X,Y \,|\, \mathbb{G}\backslash G_j)}, G_j) = 0, \tag{10}$$

*then $\mathbb{G}$ is a proper valid IV set for one of the causal relationships, i.e., $\mathbb{G} \subseteq \mathbf{G}_{\mathcal{V}}^{X \to Y}$ or $\mathbb{G} \subseteq \mathbf{G}_{\mathcal{V}}^{Y \to X}$.*

Proposition 3 indicates that, under Assumptions 1 and 2, we are able to identify a subset that consists of valid IVs for one of the relationships, from the candidate genetic variants **G**. Here, it should be noted that in a bi-directional MR model (with both $\beta_{X \to Y} \neq 0$ and $\beta_{Y \to X} \neq 0$), one cannot determine whether the identified IV set $\mathbb{G}$ is related to the causal relationship $X \to Y$ or $Y \to X$. To address this, we here introduce the following conditions that allow the estimated valid IV sets to determine the direction.

**Assumption 3.** *In a bi-directional MR model, for a given causal relationship $X \to Y$, $\beta_{X \to Y}^2 \cdot Var(X) < Var(Y)$. Similarly, for a given causal relationship $Y \to X$, $\beta_{Y \to X}^2 \cdot Var(Y) \leq Var(X)$.*

Assumption 3 is a very natural condition that one expects to hold for the unique identifiability of valid IVs for the bi-directional model. This assumption is often reasonable in MR models as discussed in Xue and Pan [2020]. One may refer to Pages 4~5 in Xue and Pan [2020] for more details.

**Proposition 4** (Identifying Direction of Causal Influences). *Suppose Assumption 3 holds. Given a valid IV set denoted as $\mathbb{G}$, if each $G_j \in \mathbb{G}$ satisfies the condition $|\text{corr}(G_j, Y)/\text{corr}(G_j, X)| < 1$, then $\mathbb{G}$ is a valid IV set for the causal relationship $X \to Y$, i.e., $\mathbb{G} \subseteq \mathbf{G}_{\mathcal{V}}^{X \to Y}$. Conversely, if this condition is not satisfied, $\mathbb{G}$ is a valid IV set for the causal relationship $Y \to X$, i.e., $\mathbb{G} \subseteq \mathbf{G}_{\mathcal{V}}^{Y \to X}$.*

Proposition 4 indicates that, under Assumption 3, it is possible to ascertain the directional orientation of the identified IV sets. Thus, with Propositions 2~4, we have sufficient and necessary conditions under which the bi-directional MR model is fully identifiable, formalized as follows:

**Theorem 1** (Identifiability of Bi-directional MR model). *Suppose that Assumptions $1 \sim 3$, and the size of participants $n \to \infty$ hold, the Bi-directional MR model is fully identifiable, which includes the identification of valid IV sets, the causal relationships the identified sets relate to, the causal directions between $X$ and $Y$, and the causal effects of $X$ on $Y$ as well as of $Y$ on $X$.*

## 4 PReBiM Algorithm

In this section, we leverage the above theoretical results and propose a data-driven algorithm, PReBiM (**P**seudo-**Re**sidual-based **Bi**-directional **M**R), for estimating the causal effects of interest. It contains two key steps, finding valid IV sets (Step I), and inferring causal direction and estimating the causal effects given the identified valid IV sets (Step II). The algorithm is designed with the following rules, which are proved correct: (1) valid IV sets can be correctly discovered given the measured genetic variants (Section 4.1), and (2) the causal direction and causal effects can be uniquely identified (Section 4.2). The entire process is summarized in Algorithm 1.

---

**Algorithm 1** PReBiM

---

**Input:** A dataset of measured genetic variants $\mathbf{G} = (G_1, ..., G_g)^{\mathsf{T}}$, two phenotypes $X$ and $Y$, significance level $\alpha$, and parameter $W$, maximum number of IVs to consider.
 1: Valid IV sets $\mathcal{V} \leftarrow \text{FindValidIVSets}(\mathbf{G}, X, Y, \alpha, W)$;  ▷ *Step I*
 2: $(\hat{\beta}_{X \to Y}, \hat{\beta}_{Y \to X}) \leftarrow \text{InferCausalDirectionEffects}(X, Y, \mathcal{V})$;  ▷ *Step II*
**Output:** $\hat{\beta}_{X \to Y}$ and $\hat{\beta}_{Y \to X}$, the causal effects of $X$ on $Y$ and $Y$ on $X$, respectively.

---

## 4.1 Step I: Finding Valid IV Sets

Propositions 2 and 3 have paved the way to distinguish the invalid and valid IV sets, respectively. This enables us to use a cluster fusion-like method to discover the valid IV sets[4]. To improve the algorithm's effectiveness with a finite sample size, especially when the pool of candidate genetic variants $\mathbf{G}$ is substantial, we introduce a parameter $W$, representing the maximum number of instrumental variables (IVs) to consider. This approach aims to circumvent the necessity of identifying the largest IV set within the collection, thereby minimizing the need for IV validity testing.

The basic process is provided in Algorithm 2. Due to limited space, the specific details of algorithm execution are provided in Appendix E. In practice, we check for the valid IV test, i.e., Eq.(10), with the Pearson Correlation Test using the Fisher transformation [Anderson, 2003], denoted by PCT.

---

**Algorithm 2** FindValidIVSets

**Input:** A dataset of measured genetic variants $\mathbf{G} = (G_1, ..., G_g)^{\mathsf{T}}$, two phenotypes $X$ and $Y$, significance level $\alpha$, and parameter $W$, maximum number of IVs to consider.

1. Initialize valid IV sets $\mathcal{V} = \emptyset$ and $\tilde{\mathbf{G}} = \mathbf{G}$.
2. Find $\mathcal{V}_{new}$ with $|\mathbb{G}| = 2$ according to minimize the sum of correlation between pseudo-residual $\mathcal{PR}_{(X,Y \,|\, \mathbb{G} \setminus G_i)}$ and $G_i$, where $G_i \in \mathbb{G}$ and $\mathbb{G} \in \tilde{\mathbf{G}}$.
3. Update $\mathcal{V}_{\text{new}}$ by incrementally adding genetic variant $G_k$ until it is impossible to add variables without passing PCT or the length of the set $\mathcal{V}_{\text{new}}$ reaches our predetermined threshold $W$.
4. Repeat Step 1 and Step 2 to find another $\mathcal{V}'_{\text{new}}$. If $\mathcal{V}$ is empty, add $\mathcal{V}'_{\text{new}}$ to $\mathcal{V}$. If not, and the combined set $\{\mathcal{V} \cup \mathcal{V}'_{\text{new}}\}$ passes the PCT, merge them; otherwise, discard $\mathcal{V}'_{\text{new}}$. Stop the iteration when $\mathcal{V}$ contains two sets of IVs or when there is one or fewer genetic variants left in $\tilde{\mathbf{G}}$.

**Output:** $\mathcal{V}$, a set that collects the valid IV sets.

---

## 4.2 Step II: Inferring Causal Direction and Estimating the Causal Effects

In this section, we infer the causal direction and estimate the causal effects given the identified valid IV sets. The basic process is provided in Algorithm 3. Due to limited space, the specific details of algorithm execution are provided in Appendix F.

---

**Algorithm 3** InferCausalDirectionEffects

**Input:** Two phenotypes $X$ and $Y$, Valid IV sets $\mathcal{V}$.

1. Infer the causal direction corresponding to the valid IV set $\mathcal{V}$ based on Proposition 4.
2. Calculate the final causal effects $\hat{\beta}_{X \to Y}$ and $\hat{\beta}_{Y \to X}$ by $\text{TSLS}(\cdot)$.

**Output:** $\hat{\beta}_{X \to Y}$ and $\hat{\beta}_{Y \to X}$

---

## 4.3 Correctness of PReBiM Algorithm

In this section, we show that, in the large sample limit, for the causal relationships of interest $X \leftrightarrow Y$, the causal effects $\hat{\beta}_{X \to Y}$ and $\hat{\beta}_{Y \to X}$ obtained through the proposed method are unbiased.

**Theorem 2** (Correctness). *Assume that the data $\{X, Y\}$ and $\mathbf{G}$ strictly follow Eq.(1) and Assumptions $1 \sim 3$ hold. Given infinite samples, the PReBiM algorithm outputs the true causal effects $\beta_{X \to Y}$ and $\beta_{Y \to X}$ correctly.*

## 4.4 Complexity of PReBiM Algorithm

We give the computational complexity analysis as follows. First, the complexity of finding a valid IV set $\mathcal{V}_{new}$ of length 2 is $\mathcal{O}\left(2\binom{g}{2}\right)$, where $g$ is the number of IVs. Then, the complexity of expanding $\mathcal{V}_{new}$ to the length $W$ is $\mathcal{O}\left(\frac{(2g-W-2)(W-1)}{2}\right)$, where $W$ is the maximum length of the IV set. For finding additional valid IV sets, the number of loops, $t$, must satisfy $t \leq \frac{g-4}{W}$. The complexity of finding the second valid IV set of length $W$ is given by $\mathcal{O}\left(\sum_{k=1}^{t}\left(2\binom{g-kW}{2} + \frac{(2g-(2k+1)W-2)(W-1)}{2}\right) + 2W(t-1) + 1\right)$.

---

[4]A cluster is considered a valid IV set, and the term "fusion-like" refers to the specific process of identifying and merging these clusters.

Therefore, the total computational complexity of the PReBiM algorithm is
$\mathcal{O}\left(\sum_{k=0}^{t}\left(2\binom{g-kW}{2}+\frac{(2g-(2k+1)W-2)(W-1)}{2}\right)+2W(t-1)+1\right)$.

## 5 Discussion

The preceding sections demonstrate that a valid IV set can be identified when any two genetic variants in $\mathbf{G}$ are independent. In this section, we explore whether the main results of Propositions 2 and 3 remain applicable when some genetic variants in $\mathbf{G}$ are dependent. As shown in Figure 4 (b), we evaluate the condition of Proposition 2 and find that $\text{corr}(\mathcal{PR}_{(X,Y,|G_2)}, G_1) \neq 0$ and $\text{corr}(\mathcal{PR}_{(X,Y,|G_1)}, G_2) \neq 0$. This indicates that $\{G_1, G_2\}$ is an invalid IV set. Conversely, in Figure 4 (a), when assessing the condition of Proposition

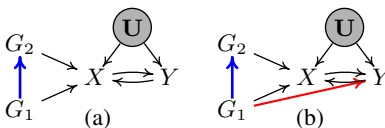

Figure 4: Simple examples, where $\{G_1, G_2\}$ is a valid IV set in (a), whereas invalid in (b).

3, we observe that $\text{corr}(\mathcal{PR}_{(X,Y,|G_2)}, G_1) = 0$ and $\text{corr}(\mathcal{PR}_{(X,Y,|G_1)}, G_2) = 0$, suggesting that $\{G_1, G_2\}$ is a valid IV set. Consequently, it is interesting to note that even with such dependence between genetic variants, our main results may still be effective in identifying valid IV sets. In Appendix H.1 $\sim$ H.2, we further demonstrate the practicality of our method for this example. As for the complex scenarios where there is confounding among genetic variants or between the genetic variants and some phenotype, or where there exist genetic variants that are associated with another phenotype while this phenotype causes $X$ and $Y$, we leave them into our future work.

## 6 Experimental Results

In this section, we conduct various simulation studies to evaluate the performance of the proposed PReBiM method. The methods we compare against are: 1) NAIVE, the least-squares regression method; 2) MR-Egger [Bowden et al., 2015]; 3) sisVIVE algorithm [Kang et al., 2016] [5]; 4) IV-TETRAD algorithm[Silva and Shimizu, 2017] [6]; 5) TSHT algorithm [Guo et al., 2018] [7]; and 6) PReBiM, our method. The second method, MR-Egger, is specifically for MR studies with invalid IVs. The last four methods are capable of handling the one-sample MR model in the presence of unmeasured confounders. Our source code can be found in the Supplementary Materials.

**Experimental setup.** To compare the performance of these methods in a realistic setting, analogous to Slob and Burgess [2020], the genetic variants are modeled as Single Nucleotide Polymorphisms (SNPs), with a varying minor allele frequency $maf_j$, and take values 0, 1, or 2. The minor allele frequencies are drawn from a uniform distribution. Specifically, the data generation process for the bi-directional model is as follows:

$$U = \mathbf{G}^\mathsf{T}\gamma_U + \varepsilon_1, \tag{11}$$
$$X = Y\beta_{Y \to X} + \mathbf{G}^\mathsf{T}\gamma_X + U\gamma_{X,U} + \varepsilon_2, \tag{12}$$
$$Y = X\beta_{X \to Y} + \mathbf{G}^\mathsf{T}\gamma_Y + U\gamma_{Y,U} + \varepsilon_3, \tag{13}$$
$$G_{ij} \sim Binomial(2, maf_j), maf_j \sim \mathcal{U}(0.1, 0.5), \tag{14}$$

where the error terms $\varepsilon_1, \varepsilon_2, \varepsilon_3$ each follow an independent normal distribution with mean 0 and unit variance. The causal effects $\beta_{Y \to X}$ and $\beta_{X \to Y}$ are generated from a uniform distribution between $[-1, -0.5] \cup [0.5, 1]$. Note that by setting $\beta_{Y \to X} = 0$ in Eq.(12), the data generation process for the one directional model can be obtained.

We here conducted experiments with the following tasks.

- **T1: Sensitivity to Sample Size.** We evaluated the impact of different sample sizes: $n = 2k, 5k$, and $10k$, where $k$ equals $1,000$.

---

[5]For sisVIVE algorithm, we used the implementations in the R sisVIVE package, which can be downloaded at https://cran.r-project.org/web/packages/sisVIVE/.

[6]For IV-TETRAD method, we used the implementations in the R package, which can be downloaded at https://www.homepages.ucl.ac.uk/~ucgtrbd/code/iv_discovery/.

[7]For the TSHT algorithm, we used the implementations in the R RobustIV package, which can be downloaded at https://cran.r-project.org/web/packages/RobustIV/.

- **T2: Sensitivity to the Number of Valid IVs.** We explored three distinct IV combination scenarios: $\mathcal{S}(2,2,6)$, $\mathcal{S}(3,3,8)$, and $\mathcal{S}(4,4,10)$. The first two elements in the parentheses represent the number of valid instrumental variables for causal relationships $X \to Y$ and $Y \to X$, respectively. The last element in the parentheses represents the total number of genetic variants.

For a comprehensive analysis, we examined the data from the Bi-directional MR model and the One-directional MR model, as detailed in Section 6.1 and 6.2, respectively. For each experimental setting, we repeated the simulation 500 times and averaged the results.

**Metrics:** We adopted the following evaluation metrics to evaluate the results of all methods.

- **(i) Correct-Selecting Rate (CSR)**: the number of correctly identified valid IVs divided by the total number of genetic variants, for the direction $X \to Y$ or $Y \to X$.
- **(ii) Mean Squared Error (MSE)**: The squared difference between the estimated causal effect and the true causal effect.

To further validate the effectiveness and practicality of our algorithm, we have expanded our experimental settings to include additional scenarios such as small sample sizes, different numbers of valid IVs, large-scale datasets, and cases where all IVs are valid. Within these settings, we compare our algorithm with the five aforementioned methods to assess algorithm performance. Detailed experimental results are provided in Appendix H.3 ∼ H.6.

## 6.1 Evaluation on Bi-Directional MR Model

In this section, we evaluate the performance of six methods on bi-directional MR data, i.e., $\beta_{X \to Y} \neq 0$ and $\beta_{Y \to X} \neq 0$. Table 1 summarizes the results of metrics CSR and MSE in a bi-directional MR model with various sample sizes. It is noted that sisVIVE, IV-TETRAD, and TSHT are not specifically designed for the bi-directional model, so we run them in both forward and reverse directions to obtain the corresponding results. As expected, our proposed PReBiM method gives the best results in all three scenarios, with all sample sizes, indicating that it can identify valid IVs and derive consistent causal effects from purely observational data, even without knowledge of the causal direction between $X$ and $Y$. Furthermore, the performance of the other three comparison methods of selecting IVs was poor, indicating that they are not suitable for bi-directional MR models. The MSE of the NAIVE method does not tend toward 0, indicating that if unobserved confounding is not considered, the estimated causal effect is certainly biased. Similarly, the MSE of MR-Egger is poor if estimated with all instrumental variables.

Table 1: Performance comparison of NAIVE, MR-Egger, sisVIVE, IV-TETRAD, TSHT, and PReBiM in estimating bi-directional MR models across various sample sizes and three scenarios.

| Size | Algorithm | $\mathcal{S}(2,2,6)$ $X \to Y$ CSR↑ | MSE↓ | $Y \to X$ CSR↑ | MSE↓ | $\mathcal{S}(3,3,8)$ $X \to Y$ CSR↑ | MSE↓ | $Y \to X$ CSR↑ | MSE↓ | $\mathcal{S}(4,4,10)$ $X \to Y$ CSR↑ | MSE↓ | $Y \to X$ CSR↑ | MSE↓ |
|---|---|---|---|---|---|---|---|---|---|---|---|---|---|
| | NAIVE | - | 0.354 | - | 0.349 | - | 0.351 | - | 0.320 | - | 0.335 | - | 0.304 |
| | MR-Egger | - | 0.800 | - | 0.734 | - | 0.571 | - | 0.569 | - | 0.457 | - | 0.494 |
| 2k | sisVIVE | 0.30 | 0.398 | 0.25 | 0.447 | 0.34 | 0.379 | 0.32 | 0.379 | 0.37 | 0.346 | 0.38 | 0.330 |
| | IV-TETRAD | 0.60 | 0.859 | 0.30 | 0.867 | 0.60 | 0.773 | 0.28 | 0.782 | 0.67 | 0.688 | 0.23 | 0.622 |
| | TSHT | 0.07 | 0.606 | 0.06 | 0.660 | 0.07 | 0.549 | 0.07 | 0.613 | 0.12 | 0.653 | 0.11 | 0.689 |
| | PReBiM | **0.85** | **0.046** | **0.85** | **0.070** | **0.89** | **0.083** | **0.87** | **0.075** | **0.88** | **0.039** | **0.89** | **0.057** |
| | NAIVE | - | 0.350 | - | 0.339 | - | 0.347 | - | 0.319 | - | 0.357 | - | 0.306 |
| | MR-Egger | - | 0.737 | - | 0.836 | - | 0.604 | - | 0.508 | - | 0.515 | - | 0.520 |
| 5k | sisVIVE | 0.32 | 0.421 | 0.30 | 0.407 | 0.33 | 0.378 | 0.35 | 0.384 | 0.39 | 0.439 | 0.38 | 0.365 |
| | IV-TETRAD | 0.62 | 0.804 | 0.31 | 0.844 | 0.63 | 0.836 | 0.29 | 0.659 | 0.67 | 0.686 | 0.25 | 0.640 |
| | TSHT | 0.04 | 0.544 | 0.06 | 0.565 | 0.08 | 0.528 | 0.06 | 0.570 | 0.10 | 0.549 | 0.07 | 0.541 |
| | PReBiM | **0.93** | **0.020** | **0.91** | **0.027** | **0.90** | **0.019** | **0.92** | **0.020** | **0.90** | **0.009** | **0.91** | **0.010** |
| | NAIVE | - | 0.366 | - | 0.348 | - | 0.319 | - | 0.342 | - | 0.349 | - | 0.300 |
| | MR-Egger | - | 0.800 | - | 0.763 | - | 0.524 | - | 0.620 | - | 0.487 | - | 0.424 |
| 10k | sisVIVE | 0.31 | 0.457 | 0.29 | 0.503 | 0.37 | 0.383 | 0.31 | 0.394 | 0.41 | 0.399 | 0.41 | 0.328 |
| | IV-TETRAD | 0.64 | 0.811 | 0.31 | 0.804 | 0.64 | 0.748 | 0.29 | 0.696 | 0.71 | 0.635 | 0.24 | 0.577 |
| | TSHT | 0.04 | 0.449 | 0.02 | 0.475 | 0.05 | 0.525 | 0.05 | 0.466 | 0.05 | 0.479 | 0.07 | 0.498 |
| | PReBiM | **0.95** | **0.044** | **0.93** | **0.018** | **0.93** | **0.026** | **0.93** | **0.039** | **0.93** | **0.027** | **0.94** | **0.011** |

Note: The symbol "−" means methods have no output. "↑" means a higher value is better, and vice versa.

## 6.2 Evaluation on One-Directional MR Model

In this section, we evaluate the performance of six methods on one-directional MR data, e.g., $\beta_{X \to Y} \neq 0$ and $\beta_{Y \to X} = 0$. For fairness, we assume that the causal direction $X \to Y$ is known for all methods. Figure 5 summarizes the results of metrics CSR (solid lines) and MSE (dashed

lines) for all methods in a one-directional MR model with different scenarios. As expected, the metrics CSR and MSE of our method approach the expected value as the sample size increases. The overall results are comparable to the IV-TETRAD algorithm and superior to the sisVIVE algorithm and TSHT algorithm. This is because our method, similar to the IV-TETRAD method, requires fewer valid IVs in the system (at least 2), whereas the other two methods require the number of valid IVs that are related to the overall genetic variants, namely the majority rule assumption (over $50\%$ of the candidate IVs are valid) and plurality rule assumption (the number of valid IVs is larger than any number of invalid IVs). It has been noted that under the IV-TETRAD method for scenario $\mathcal{S}(2,0,6)$, the metrics CSR and MSE outperform our method. This is because the IV-TETRAD method requires a predetermined number of IV sets to be specified in advance, and in this case, we have predetermined the number of valid IVs for it. However, our method does not have this prior information.

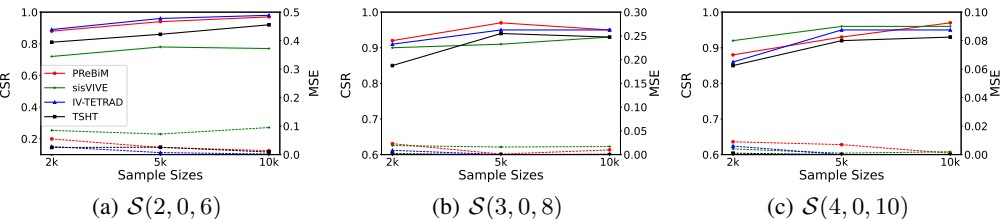

(a) $\mathcal{S}(2,0,6)$       (b) $\mathcal{S}(3,0,8)$       (c) $\mathcal{S}(4,0,10)$

Figure 5: Performance comparison of sisVIVE, IV-TETRAD, TSHT, and PReBiM in estimating one-directional MR models across various sample sizes and three scenarios.

## 6.3 Evaluation on Real-World Data

We first apply our method to analyze the bi-directional causal relationships between obesity and vitamin D status using the GWAS data from Vimaleswaran et al. [2013]. Note that the original data only consists of summary data, and thus we generate individual-level data based on the data generation mechanism described in Eq.(1). The data we used comprises 16 related SNPs as candidate IVs, including seven SNPs associated with obesity (BMI): rs9939609 (FTO), rs2867125 (TMEM18), rs4074134 (BDNF), rs7647305 (ETV5), rs7138803 (FAIM2), rs3101336 (NEGR1), and rs10938397 (GNPDA2); four SNPs related to 25-hydroxyvitamin D (25(OH)D): rs12785878 (DHCR7), rs10741657 (CYP2R1), rs2282679 (GC), and rs6013897 (CYP24A1); and five SNPs associated with BMI and 25(OH)D: rs4498364, rs1381660, rs4549702, rs12760109, and rs77913838. We observed that FTO and FAIM2 serve as valid IVs for the $X \to Y$, while DHCR7 and CYP24A1 are valid IVs for the $Y \to X$, with estimated causal effects of -1.15 and -0.05, respectively. These findings are consistent with those findings in [Vimaleswaran et al., 2013].

Next, we apply our method to analyze the causal effect of institutions on economic development using the Colonial Origins dataset from [Acemoglu et al., 2001]. The data includes eight candidate IVs: absolute latitude (lat_abst), the density of European settlements in 1900 (euro1900), the level of democracy in 1900 (democ1), constitutional constraints in 1900 (cons1), the timing of industrialization (indtime), democracy in 2000 (democ00a), constitutional constraints in 2000 (cons00a), and log-transformed European settler mortality rates (logem4). Our method identifies euro1900 and logem4 as valid IVs, with an estimated causal effect of 0.861, which is consistent with the findings reported in Acemoglu et al. [2001].

## 7 Conclusion

We investigated the identifiability conditions for the bi-directional MR model using purely observational data. First, we proposed a testable constraint for identifying valid IV sets, and we demonstrated that, under appropriate conditions, these valid IV sets can be partially or even fully identified by checking the proposed constraints. Experimental results on both datasets further verified the effectiveness of our algorithm. Currently, we assume that genetic variants are randomized. Allowing for general dependent genetic variants is a direction for future work. Additionally, while we focus on a linear bi-directional MR model, in future research, we will generalize to address nonlinear or even more complex data.

## Acknowledgements

Feng Xie would like to acknowledge the support by the National Natural Science Foundation of China (62306019). Yan Zeng would like to acknowledge the support of the Beijing Municipal Education Commission Science and Technology Program General Project (KM202410011016). Feng Xie and Yan Zeng acknowledge the support by the funding from the Disciplinary funding of Beijing Technology and Business University, the Beijing Municipal Education Commission for the Emerging Interdisciplinary Platform for Digital Business at Beijing Technology and Business University, and the Beijing Key Laboratory of Applied Statistics and Digital Regulation. Zhen Yao acknowledges the support of the Graduate Research Ability Enhancement Program Project Funding at Beijing Technology and Business University in 2024. We appreciate the constructive comments from all anonymous reviewers, which greatly helped to improve this paper.

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

# A   More Details on MR

The basic goal of epidemiologic studies is to assess the effect of changes in exposure on outcomes. The causal effect of exposure on outcome is often different from the observed correlation due to confounding factors. Correlations between exposure and outcome cannot be used as reliable evidence for inferring causality. In contrast, **Mendelian Randomization** (MR) studies use genetic variants to infer the causal effect of exposure on outcome. The idea behind MR studies is to find a genetic variant (or multiple genetic variants) that is associated with exposure but not with other confounders and that does not directly affect the outcome. Such a genetic variant in principle fulfills the basic assumptions of instrumental variables (IVs). It is then used to assess the causal effect of the exposure on the outcome [Thomas and Conti, 2004]. Since the assumptions of IVs cannot be fully tested and may be violated during IVs' selection, a number of methods have emerged to identify valid IVs.

MR studies can be conducted using a single sample, i.e., **one-sample Mendelian Randomization**, in which individuals are tested for genetic variation, exposure, and outcome in the same population. On the contrary, in **two-sample Mendelian Randomization**, the association between genetic variation and exposure is estimated in one dataset while that between genetic variation and outcome is estimated in another dataset. Compared with the Two-sample MR, one-sample MR provides more direct evidence by eliminating potential biases that can arise from individual-level data, which turns out to be more meaningful yet challenging.

Additionally, most existing MR methods assume a one-directional causal relationship between exposure and outcome, whereas bi-directional relationships are ubiquitous in real-life scenarios. For instance, there exist bi-directional relationships between obesity and vitamin D status [Vimaleswaran et al., 2013], body mass index (BMI) and type 2 diabetes (T2D), diastolic blood pressure and stroke [Xue and Pan, 2022], insomnia and five major psychiatric disorders [Gao et al., 2019], smoking and BMI [Carreras-Torres et al., 2018], etc. It is thus desirable to take further research on bi-directional MR. **Bi-directional Mendelian Randomization** assesses not only the causal effect of the exposure on the outcome but also the effect of the outcome on the exposure. To achieve this, valid IVs for both the exposure and the outcome are needed. However, this is tough because genetic variants associated with the exposure are often also associated with the outcome, and vice versa. Our goal here is to identify and differentiate valid IVs for each direction within a one-sample MR framework and infer bi-directional causal effects.

Regarding our bi-directional MR causal model, we give further relative descriptions in the following. **(i)** Similar to Li and Ye [2022], our model in Eq.(1) describes bi-directional causality in equilibrium, and in econometrics, it is known as the simultaneous equation model. According to Hausman [1983], to avoid the recursive formalization, Eq.(1) equivalently is transformed into Eq.(2) with $\beta_{X \to Y} \beta_{Y \to X} \neq 1$. **(ii)** For such a bi-directional (cyclic) structure, Mooij et al. [2013] take an Ordinary Differential Equation (ODEs) perspective and show under which conditions the equilibrium state of a system of the first-order ODEs can be described by a deterministic structural causal model (SCM). This is a very interesting work. Please note that our setting is different from those in Mooij et al. [2013]: our model is motivated by bi-directional MR studies, other than a dynamical system from the control field. Further, due to the specificity of MR studies, we can only obtain the snapshot data or equilibrium data, other than dynamical ones for the treatment, the outcome as well as genetic variants.

# B    Notation List

<p align="center">Table 2: List of major notations with descriptions.</p>

| NOTATION | DESCRIPTION |
|---|---|
| $\mathbf{U}$ | Unmeasured Confounders |
| $\mathbf{G}$ | Genetic variants |
| $\mathbb{G} \subset \mathbf{G}$ | Candidate genetic variants |
| $X$ | Exposure or treatment |
| $Y$ | Outcome |
| $n$ | Sample size |
| $\boldsymbol{\gamma}_X, \boldsymbol{\gamma}_Y, \boldsymbol{\gamma}_U$ | Effects of $\mathbf{G}$ on $X$, $\mathbf{G}$ on $Y$, and of $\mathbf{G}$ on $U$, respectively |
| $\Delta$ | A reduction factor calculated with $1 - \beta_{X \to Y}\beta_{Y \to X}$ |
| $\varepsilon_X, \varepsilon_Y$ | Error terms of $X$, and of $Y$, respectively |
| $\beta_{X \to Y}, \beta_{Y \to X}$ | True effects of $X$ on $Y$, and of $Y$ on $X$, respectively |
| $\hat{\beta}_{X \to Y}, \hat{\beta}_{Y \to X}$ | Estimated causal effects from candidate IVs for the directions $X \to Y$ and $Y \to X$, respectively |
| $\mathbf{G}_{\mathcal{V}}^{X \to Y}, \mathbf{G}_{\mathcal{V}}^{Y \to X}$ | A set consisting of all valid IVs in $\mathbf{G}$ relative to $X \to Y$, and $Y \to X$, respectively |
| $\mathbf{G}_{\mathcal{I}}^{X \to Y}, \mathbf{G}_{\mathcal{I}}^{Y \to X}$ | A set consisting at least one invalid IV in $\mathbf{G}$ relative to $X \to Y$ or $Y \to X$, respectively |
| $\text{TSLS}(\cdot)$ | The procedure of estimating causal effects using two-stage least square method |
| $\omega_{\mathbb{G}}$ | Estimated causal effect derived from $\mathbb{G}$ |
| $\mathcal{PR}_{(X,Y \mid \mathbb{G} \backslash G_j)}$ | Pseudo-residual of $\{X, Y\}$ relative to $\mathbb{G} \backslash G_j$ |
| $PCT$ | Pearson Correlation Test |

# C    More Details on Observations 1 and 2

## C.1    Observation 1

Since $\mathbf{G}_{\mathcal{V}}^{X \to Y} = (G_1, G_3)^\intercal$ is a valid IV set, employing $G_1$ or $G_3$ in $\text{TSLS}(\cdot)$ entails unbiased causal effects, i.e., $\omega_{\{G_i\}} = \text{TSLS}(X, Y, \{G_i\}) = \beta_{X \to Y}$ with $i \in \{1, 3\}$. Thus, we have,

$$Y - X\omega_{\{G_1\}} = Y - X\beta_{X \to Y} = G_2 \gamma_{Y,2} + G_4 \gamma_{Y,4} + G_5 \gamma_{Y,5} + \varepsilon_Y,$$
$$Y - X\omega_{\{G_3\}} = Y - X\beta_{X \to Y} = G_2 \gamma_{Y,2} + G_4 \gamma_{Y,4} + G_5 \gamma_{Y,5} + \varepsilon_Y.$$

Because both $G_1$ and $G_3$ are valid IVs, they are independent of $\varepsilon_Y$; and any two genetic variants are assumed to be independent of each other, we get,

$$\text{corr}(Y - X\omega_{\{G_3\}}, G_1) = 0,$$
$$\text{corr}(Y - X\omega_{\{G_1\}}, G_3) = 0.$$

## C.2    Observation 2

Without loss of generality, we here give details about how $\text{corr}(Y - X\omega_{\{G_4, G_5\}}, G_2) \neq 0$ is derived. The other two inequalities could be obtained in the same way. Following the procedure of $\text{TSLS}(\cdot)$, we first get the estimate of $X$ with $\{G_4, G_5\}$, denoted by $\hat{X}$, and obtain $\omega_{\{G_4, G_5\}}$ using $\hat{X}$ and $Y$.

$$\hat{X} = G_4 \hat{\beta}_4 + G_5 \hat{\beta}_5$$
$$= G_4(\gamma_{X,4} + \gamma_{Y,4}\beta_{Y \to X})\Delta + G_5(\gamma_{X,5} + \gamma_{Y,5}\beta_{Y \to X})\Delta$$
$$= (G_4 \gamma_{X,4} + G_4 \gamma_{Y,4}\beta_{Y \to X} + G_5 \gamma_{X,5} + G_5 \gamma_{Y,5}\beta_{Y \to X})\Delta,$$

where since $G_4$ and $G_5$ are neither correlated with the latent confounder $U$ nor the other IVs as shown in Figure 2, the second equality holds. Then, we could get the estimate of $\omega_{\{G_4, G_5\}}$, i.e.,

$$\omega_{\{G_4, G_5\}} = \frac{cov(G_4, G_4)(\gamma_{X,4} + \gamma_{Y,4}\beta_{Y \to X})(\gamma_{Y,4} + \gamma_{X,4}\beta_{X \to Y})}{cov(G_4, G_4)(\gamma_{X,4} + \gamma_{Y,4}\beta_{Y \to X})^2 + cov(G_5, G_5)(\gamma_{X,5} + \gamma_{Y,5}\beta_{Y \to X})^2}$$
$$+ \frac{cov(G_5, G_5)(\gamma_{X,5} + \gamma_{Y,5}\beta_{Y \to X})(\gamma_{Y,5} + \gamma_{X,5}\beta_{X \to Y})}{cov(G_4, G_4)(\gamma_{X,4} + \gamma_{Y,4}\beta_{Y \to X})^2 + cov(G_5, G_5)(\gamma_{X,5} + \gamma_{Y,5}\beta_{Y \to X})^2}.$$

Then we get,

$$\text{corr}(Y - X\omega_{\{G_4,G_5\}}, G_2)$$
$$= (\gamma_{Y,2} + \gamma_{X,2}\beta_{X\to Y})\Delta - (\gamma_{X,2} + \gamma_{Y,2}\beta_{Y\to X})\Delta\omega_{\{G_4,G_5\}}$$
$$= [\gamma_{X,2}(\beta_{X\to Y} - \omega_{\{G_4,G_5\}}) + \gamma_{Y,2}(1 - \beta_{Y\to X}\omega_{\{G_4,G_5\}})]\Delta.$$

Since,

$$\beta_{X\to Y} - \omega_{\{G_4,G_5\}}$$
$$= \frac{\beta_{X\to Y}(\gamma_{X,4} + \gamma_{Y,4}\beta_{Y\to X})^2 cov(G_4,G_4) + \beta_{X\to Y}(\gamma_{X,5} + \gamma_{Y,5}\beta_{Y\to X})^2 cov(G_5,G_5)}{(\gamma_{X,4} + \gamma_{Y,4}\beta_{Y\to X})^2 cov(G_4,G_4) + (\gamma_{X,5} + \gamma_{Y,5}\beta_{Y\to X})^2 cov(G_5,G_5)}$$
$$- \frac{(\gamma_{X,4} + \gamma_{Y,4}\beta_{Y\to X})(\gamma_{Y,4} + \gamma_{X,4}\beta_{X\to Y})cov(G_4,G_4)}{(\gamma_{X,4} + \gamma_{Y,4}\beta_{Y\to X})^2 cov(G_4,G_4) + (\gamma_{X,5} + \gamma_{Y,5}\beta_{Y\to X})^2 cov(G_5,G_5)}$$
$$- \frac{(\gamma_{X,5} + \gamma_{Y,5}\beta_{Y\to X})(\gamma_{Y,5} + \gamma_{X,5}\beta_{X\to Y})cov(G_5,G_5)}{(\gamma_{X,4} + \gamma_{Y,4}\beta_{Y\to X})^2 cov(G_4,G_4) + (\gamma_{X,5} + \gamma_{Y,5}\beta_{Y\to X})^2 cov(G_5,G_5)}$$
$$= \frac{(\beta_{X\to Y}\beta_{Y\to X} - 1)[\gamma_{Y,4}(\gamma_{X,4} + \gamma_{Y,4}\beta_{Y\to X}) + \gamma_{Y,5}(\gamma_{X,5} + \gamma_{Y,5}\beta_{Y\to X})]}{(\gamma_{X,4} + \gamma_{Y,4}\beta_{Y\to X})^2 + (\gamma_{X,5} + \gamma_{Y,5}\beta_{Y\to X})^2}$$
$$= \frac{(\beta_{X\to Y}\beta_{Y\to X} - 1)(\mathbf{C}_{44}\Gamma_{X,4}\gamma_{Y,4} + \mathbf{C}_{55}\Gamma_{X,5}\gamma_{Y,5})}{\mathbf{C}_{44}\Gamma_{X,4}^2 + \mathbf{C}_{55}\Gamma_{X,5}^2},$$

and

$$1 - \beta_{Y\to X}\omega_{\{G_4,G_5\}}$$
$$= \frac{(\gamma_{X,4} + \gamma_{Y,4}\beta_{Y\to X})^2 cov(G_4,G_4) + (\gamma_{X,5} + \gamma_{Y,5}\beta_{Y\to X})^2 cov(G_5,G_5)}{(\gamma_{X,4} + \gamma_{Y,4}\beta_{Y\to X})^2 cov(G_4,G_4) + (\gamma_{X,5} + \gamma_{Y,5}\beta_{Y\to X})^2 cov(G_5,G_5)}$$
$$- \frac{\beta_{Y\to X}(\gamma_{X,4} + \gamma_{Y,4}\beta_{Y\to X})(\gamma_{Y,4} + \gamma_{X,4}\beta_{X\to Y})cov(G_4,G_4)}{(\gamma_{X,4} + \gamma_{Y,4}\beta_{Y\to X})^2 cov(G_4,G_4) + (\gamma_{X,5} + \gamma_{Y,5}\beta_{Y\to X})^2 cov(G_5,G_5)}$$
$$- \frac{\beta_{Y\to X}(\gamma_{X,5} + \gamma_{Y,5}\beta_{Y\to X})(\gamma_{Y,5} + \gamma_{X,5}\beta_{X\to Y})cov(G_5,G_5)}{(\gamma_{X,4} + \gamma_{Y,4}\beta_{Y\to X})^2 cov(G_4,G_4) + (\gamma_{X,5} + \gamma_{Y,5}\beta_{Y\to X})^2 cov(G_5,G_5)}$$
$$= \frac{[\gamma_{X,4}(\gamma_{X,4} + \gamma_{Y,4}\beta_{Y\to X})cov(G_4,G_4) + \gamma_{X,5}(\gamma_{X,5} + \gamma_{Y,5}\beta_{Y\to X})cov(G_5,G_5)]\Delta}{(\gamma_{X,4} + \gamma_{Y,4}\beta_{Y\to X})^2 cov(G_4,G_4) + (\gamma_{X,5} + \gamma_{Y,5}\beta_{Y\to X})^2 cov(G_5,G_5)}$$
$$= \frac{\Delta(\mathbf{C}_{44}\Gamma_{X,4}\gamma_{X,4} + \mathbf{C}_{55}\Gamma_{X,5}\gamma_{X,5})}{\mathbf{C}_{44}\Gamma_{X,4}^2 + \mathbf{C}_{44}\Gamma_{X,5}^2},$$

Where we define $\Gamma_{X,i} = \gamma_{X,i} + \gamma_{Y,i}\beta_{Y\to X}$, $\Gamma_{Y,j} = \gamma_{Y,j} + \gamma_{X,j}\beta_{X\to Y}$, $\mathbf{C}_{ij} = cov(G_i, G_j)$ and $\Delta \triangleq 1 - \beta_{Y\to X}\beta_{X\to Y}$. we can finally obtain $\text{corr}(Y - X\omega_{\{G_4,G_5\}}, G_2)$, that is,

$$\text{corr}(Y - X\omega_{\{G_4,G_5\}}, G_2)$$
$$= \frac{\gamma_{X,2}(\beta_{X\to Y}\beta_{Y\to X} - 1)(\mathbf{C}_{44}\Gamma_{X,4}\gamma_{Y,4} + \mathbf{C}_{55}\Gamma_{X,5}\gamma_{Y,5})\Delta}{\mathbf{C}_{44}\Gamma_{X,4}^2 + \mathbf{C}_{55}\Gamma_{X,5}^2}$$
$$+ \frac{\gamma_{Y,2}(1 - \beta_{Y\to X}\beta_{X\to Y})(\mathbf{C}_{44}\Gamma_{X,4}\gamma_{X,4} + \mathbf{C}_{55}\Gamma_{X,5}\gamma_{X,5})\Delta}{\mathbf{C}_{44}\Gamma_{X,4}^2 + \mathbf{C}_{55}\Gamma_{X,5}^2}$$
$$= \frac{\mathbf{C}_{44}\Gamma_{X,4}(\gamma_{X,2}\gamma_{Y,4} - \gamma_{X,4}\gamma_{Y,2}) + \mathbf{C}_{55}\Gamma_{X,5}(\gamma_{X,2}\gamma_{Y,5} - \gamma_{X,5}\gamma_{Y,2})}{\mathbf{C}_{44}\Gamma_{X,4}^2 + \mathbf{C}_{55}\Gamma_{X,5}^2} \neq 0.$$

# D  More Details on the Example Provided in Section 5

Figure 6 gives an illustration of our idea, demonstrating that even when IVs are correlated, we can still identify them as valid IVs using our proposed method. In subgraph $(a)$, there are two valid IVs: $G_1$ and $G_2$. However, in subgraph $(b)$, these two IVs become invalid as $G_1$ is correlated with $Y$, rendering them invalid.

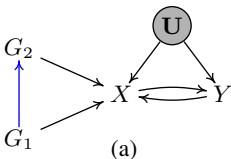 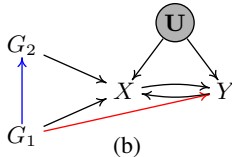

Figure 6: A simple example, where the set $\{G_1, G_2\}$ is a valid IV set in subgraph (a), whereas it is not valid in subgraph (b).

Assume the generating process of $X$ and $Y$ in subgraph $(a)$ follows:

$$
\begin{aligned}
X &= Y\beta_{Y\to X} + G_1\gamma_{X,1} + G_2\gamma_{X,2} + \varepsilon_X, \\
Y &= X\beta_{X\to Y} + \varepsilon_Y, \\
G_2 &= G_1\gamma_{1,2} + \varepsilon_2.
\end{aligned}
\tag{15}
$$

where $\varepsilon_X$ and $\varepsilon_Y$ are **correlated**, and $\varepsilon_2$ represents the independent error term.
For $G_1 \in \mathbb{G}$, we check the condition of Proposition 3:

$$
\omega_{\{G_1\}} = \mathrm{TSLS}(X, Y, \{G_1\}) = \frac{\mathrm{cov}(Y, G_1)}{\mathrm{cov}(X, G_1)} = \beta_{X\to Y} + \frac{\mathrm{cov}(\varepsilon_Y, G_1)}{\mathrm{cov}(X, G_1)} = \beta_{X\to Y}
$$

$$
\mathcal{PR}_{(X,Y\,|\,\{G_1\})} = Y - \omega_{\{G_1\}}X = \varepsilon_Y
$$

$$
\mathrm{corr}(\mathcal{PR}_{(X,Y\,|\,\{G_1\})}, G_2) = 0
$$

Similarly, for $G_2 \in \mathbb{G}$, we check the condition of Proposition 3:

$$
\omega_{\{G_2\}} = \mathrm{TSLS}(X, Y, \{G_2\}) = \frac{\mathrm{cov}(Y, G_2)}{\mathrm{cov}(X, G_2)} = \beta_{X\to Y} + \frac{\mathrm{cov}(\varepsilon_Y, G_2)}{\mathrm{cov}(X, G_2)} = \beta_{X\to Y}
$$

$$
\mathcal{PR}_{(X,Y\,|\,\{G_2\})} = Y - \omega_{\{G_2\}}X = \varepsilon_Y
$$

$$
\mathrm{corr}(\mathcal{PR}_{(X,Y\,|\,\{G_2\})}, G_1) = 0
$$

Assume the generating process of $X$ and $Y$ in subgraph $(b)$ follows Equation 16:

$$
\begin{aligned}
X &= Y\beta_{Y\to X} + G_1\gamma_{X,1} + G_2\gamma_{X,2} + \varepsilon_X, \\
Y &= X\beta_{X\to Y} + G_1\gamma_{Y,1} + \varepsilon_Y, \\
G_2 &= G_1\gamma_{1,2} + \varepsilon_2.
\end{aligned}
\tag{16}
$$

where $\varepsilon_X$ and $\varepsilon_Y$ are **correlated**, and $\varepsilon_2$ represents the independent error term.
Then, Equation 16 can be reorganized to avoid recursive formulations as follows:

$$
\begin{aligned}
X &= \Delta\left(G_1(\gamma_{X,1} + \gamma_{1,2}\gamma_{X,2} + \gamma_{Y,1}\beta_{Y\to X}) + \varepsilon_2\gamma_{X,2} + \varepsilon_X + \varepsilon_Y\beta_{Y\to X}\right) \\
&= \Delta\left(G_1\Gamma_{X,1} + \varepsilon_2\gamma_{X,2} + \varepsilon_X + \varepsilon_Y\beta_{Y\to X}\right) \\
Y &= \Delta\left(G_1(\gamma_{Y,1} + \beta_{X\to Y}(\gamma_{X,1} + \gamma_{X,2}\gamma_{1,2})) + \varepsilon_2\gamma_{X,2}\beta_{X\to Y} + \varepsilon_X\beta_{X\to Y} + \varepsilon_Y\right) \\
&= \Delta\left(G_1\Gamma_{Y,1} + \varepsilon_2\gamma_{X,2}\beta_{X\to Y} + \varepsilon_X\beta_{X\to Y} + \varepsilon_Y\right)
\end{aligned}
\tag{17}
$$

where $\Gamma_{X,1} = \gamma_{X,1} + \gamma_{1,2}\gamma_{X,2} + \gamma_{Y,1}\beta_{Y\to X}$ and $\Gamma_{Y,1} = \gamma_{Y,1} + \beta_{X\to Y}(\gamma_{X,1} + \gamma_{X,2}\gamma_{1,2})$. It is noted that $\Gamma_{X,1} \neq 0$ and $\Gamma_{Y,1} \neq 0$.
In subgraph $(b)$, for $G_2 \in \mathbb{G}$, we check the condition of Proposition 3:

$$\omega_{\{G_2\}} = \text{TSLS}(X, Y, \{G_2\}) = \frac{\text{cov}(Y, G_2)}{\text{cov}(X, G_2)} = \beta_{X \to Y} + \gamma_{Y,1} \frac{\text{cov}(G_1, G_2)}{\text{cov}(X, G_2)}$$

$$= \beta_{X \to Y} + \frac{\gamma_{Y,1}\gamma_{1,2}}{\Delta} \frac{\text{cov}(G_1, G_1)}{\text{cov}(G_1, G_1)(\gamma_{1,2}\Gamma_{X,1}) + \text{cov}(\varepsilon_2, \varepsilon_2)\gamma_{X,2}}$$

$$= \beta_{X \to Y} + \frac{\gamma_{Y,1}\gamma_{1,2}}{\Delta} \frac{1}{\gamma_{1,2}\Gamma_{X,1} + \gamma_{X,2}\mathcal{M}},$$

$$\mathcal{PR}_{(X,Y \,|\, \{G_2\})} = Y - \omega_{\{G_2\}}X$$

$$= G_1\gamma_{Y,1} + \varepsilon_Y - \frac{\gamma_{Y,1}\gamma_{1,2}}{\Delta} \frac{1}{\gamma_{1,2}\Gamma_{X,1} + \gamma_{X,2}\mathcal{M}}X$$

$$= G_1\gamma_{Y,1} + \varepsilon_Y - \frac{\gamma_{Y,1}\gamma_{1,2}(G_1\Gamma_{X,1} + \varepsilon_2\gamma_{X,2} + \varepsilon_X + \varepsilon_Y\beta_{Y \to X})}{\gamma_{1,2}\Gamma_{X,1} + \gamma_{X,2}\mathcal{M}},$$

$$\text{corr}(\mathcal{PR}_{(X,Y \,|\, \{G_2\})}, G_1) = \gamma_{Y,1} - \frac{\gamma_{Y,1}\gamma_{1,2}\Gamma_{X,1}}{\gamma_{1,2}\Gamma_{X,1} + \gamma_{X,2}\mathcal{M}}$$

$$= \frac{\gamma_{Y,1}\gamma_{1,2}\Gamma_{X,1} + \gamma_{Y,1}\gamma_{X,2}\mathcal{M} - \gamma_{Y,1}\gamma_{1,2}\Gamma_{X,1}}{\gamma_{1,2}\Gamma_{X,1} + \gamma_{X,2}\mathcal{M}}$$

$$= \frac{\gamma_{Y,1}\gamma_{X,2}\mathcal{M}}{\gamma_{1,2}\Gamma_{X,1} + \gamma_{X,2}\mathcal{M}}$$

$$(18)$$

where $\mathcal{M} = \frac{\text{cov}(\varepsilon_2, \varepsilon_2)}{\text{cov}(G_1, G_1)} \neq 0$.

Based on these two observations, we can conclude that the two IVs in subgraph $(a)$ can be deemed valid. However, in subgraph $(b)$, they can be deemed invalid.

# E More Details on FindValidIVSets Algorithm (in Section 4.1)

It is noteworthy that, to mitigate the unreliability of statistical tests in finite samples, in practice, we choose to take the smallest correlation coefficient to select a valid IV set, as in Line 4 or 10 of Algorithm 4.

---

**Algorithm 4** FindValidIVSets

---

**Input:** A dataset of measured genetic variants $\mathbf{G} = (G_1, ..., G_g)^\intercal$, two phenotypes $X$ and $Y$, significance level $\alpha$, and parameter $W$, maximum number of IVs to consider.

1: Initialize valid IV sets $\mathcal{V} = \emptyset$ and $\tilde{\mathbf{G}} = \mathbf{G}$.
2: **repeat**
3:     For all subsets $\mathbb{G} \in \tilde{\mathbf{G}}$ with $|\mathbb{G}| = 2$, compute the correlation between pseudo-residual $\mathcal{PR}_{(X,Y \,|\, \mathbb{G}\backslash G_i)}$ and $G_i$, where $G_i \in \mathbb{G}$.
4:     $\mathcal{V}_{new} \leftarrow \underset{\mathbb{G}\in\mathbf{G}}{\arg\min} \sum\limits_{i=1}^{2} \mathrm{corr}(\mathcal{PR}_{(X,Y \,|\, \mathbb{G}\backslash G_i)}, G_i)$.
5:     **repeat**
6:         $\tilde{\mathbf{G}}' \leftarrow \tilde{\mathbf{G}}$
7:         For each variable $G_k \in \tilde{\mathbf{G}}'$, remove all variables from $\tilde{\mathbf{G}}'$ corresponding to those that set $\{\mathcal{V}, G_k\}$ fail to the PCT.
8:         **if** $\tilde{\mathbf{G}}' \neq \emptyset$ **then**
9:             Compute the correlation between pseudo-residual $\mathcal{PR}_{(X,Y \,|\, \mathcal{V})}$ and $G_k$.
10:            $G' \leftarrow \underset{G_k \in \tilde{\mathbf{G}}'}{\arg\min} \mathrm{corr}(\mathcal{PR}_{(X,Y \,|\, \mathcal{V})}, G_k)$.
11:            $\mathcal{V}_{new} \leftarrow \mathcal{V}_{new} \cup \{G_k\}$.
12:         **else**
13:            Break the loop of line 5;
14:         **end if**
15:     **until** the length of the set $\mathcal{V}_{new}$ has reached our predefined threshold $W$.
16:     **if** Set $\mathcal{V} = \emptyset$ **then**
17:         Add set $\mathcal{V}_{new}$ into $\mathcal{V}$.
18:     **else**
19:         **if** Set $\{\mathcal{V} \cup \mathcal{V}_{new}\}$ fail to PCT **then**
20:            Add set $\mathcal{V}_{new}$ into $\mathcal{V}$.
21:         **else**
22:            Delete set $\mathcal{V}_{new}$ from $\tilde{\mathbf{G}}$.
23:         **end if**
24:     **end if**
25: **until** $\mathcal{V}$ contains two sets or $|\tilde{\mathbf{G}}| \leq 1$.

**Output:** $\mathcal{V}$, a set that collects the valid IV sets.

---

# F   More Details on InferCausalDirectionEffects Algorithm (in Section 4.2)

---

**Algorithm 5** InferCausalDirectionEffects

---

**Input:** Two phenotypes $X$ and $Y$, Valid IV sets $\mathcal{V}$.

1: Initialize sets $\mathbf{G}_{\mathcal{V}}^{X \to Y} = \emptyset$ and $\mathbf{G}_{\mathcal{V}}^{Y \to X} = \emptyset$, and the causal effects $\hat{\beta}_{X \to Y} = \emptyset$ and $\hat{\beta}_{Y \to X} = \emptyset$.
2: **for** each $G_j \in \mathcal{V}$ **do**
3:     **if** $|\operatorname{corr}(G_j, Y)/\operatorname{corr}(G_j, X)| < 1$ **then**
4:         Add $G_j$ into $\mathbf{G}_{\mathcal{V}}^{X \to Y}$;
5:     **else**
6:         Add $G_j$ into $\mathbf{G}_{\mathcal{V}}^{Y \to X}$;
7:     **end if**
8: **end for**
9: $\hat{\beta}_{X \to Y} \leftarrow \text{TSLS}(X, Y, \mathbf{G}_{\mathcal{V}}^{X \to Y})$.
10: $\hat{\beta}_{Y \to X} \leftarrow \text{TSLS}(Y, X, \mathbf{G}_{\mathcal{V}}^{Y \to X})$.

**Output:** $\hat{\beta}_{X \to Y}$ and $\hat{\beta}_{Y \to X}$

---

# G   Proofs

## G.1   Proof of Proposition 1

*Proof.* From procedure "two-stage least squares", we obtained the fitted values $\hat{X}$ from the regression in the first stage: $\hat{X} = (\mathbf{G}_{\mathcal{V}}^{X \to Y})^{\intercal} \hat{\alpha}$. According to OLS regression,

$$\begin{aligned}
\hat{X} &= (\mathbf{G}_{\mathcal{V}}^{X \to Y})^{\intercal} \left[ (\mathbf{G}_{\mathcal{V}}^{X \to Y} (\mathbf{G}_{\mathcal{V}}^{X \to Y})^{\intercal})^{-1} \mathbf{G}_{\mathcal{V}}^{X \to Y} X \right] \\
&= \left[ (\mathbf{G}_{\mathcal{V}}^{X \to Y})^{\intercal} (\mathbf{G}_{\mathcal{V}}^{X \to Y} (\mathbf{G}_{\mathcal{V}}^{X \to Y})^{\intercal})^{-1} \mathbf{G}_{\mathcal{V}}^{X \to Y} \right] X \\
&= \mathbf{P} X.
\end{aligned}$$

Then run the OLS regression for $Y$ on $\hat{X}$ in second stage: $Y = \hat{\beta} \hat{X}$. where:

$$\begin{aligned}
\hat{\beta} &= (\hat{X}^{\intercal} \hat{X})^{-1} (\hat{X}^{\intercal} Y) \\
&= ((\mathbf{P}X)^{\intercal} \mathbf{P}X)^{-1} ((\mathbf{P}X)^{\intercal} Y) \\
&= (X^{\intercal} \mathbf{P} X)^{-1} (X^{\intercal} \mathbf{P} Y) \\
&= (X^{\intercal} \mathbf{P} X)^{-1} (X^{\intercal} \mathbf{P}) \left( X \beta_{X \to Y} + (\mathbf{G}_{\mathcal{V}}^{X \to Y})^{\intercal} \boldsymbol{\gamma}_Y + \varepsilon_Y \right) \\
&= \beta_{X \to Y} + (X^{\intercal} \mathbf{P} X)^{-1} (X^{\intercal} \mathbf{P} (\mathbf{G}_{\mathcal{V}}^{X \to Y})^{\intercal} \boldsymbol{\gamma}_Y) + (X^{\intercal} \mathbf{P} X)^{-1} (X^{\intercal} \mathbf{P} \varepsilon_Y)
\end{aligned}$$

Since $\mathbf{G}_{\mathcal{V}}^{X \to Y}$ is a valid IV set, for $G_j \in \mathbf{G}_{\mathcal{V}}^{X \to Y}$, both $\gamma_{Y,j} = 0$ (A2) and $cov(G_j, \varepsilon_Y) = 0$ (A3). Hence, we obtained

$$(X^{\intercal} \mathbf{P} X)^{-1} (X^{\intercal} \mathbf{P} (\mathbf{G}_{\mathcal{V}}^{X \to Y})^{\intercal} \boldsymbol{\gamma}_Y) = 0$$
$$(X^{\intercal} \mathbf{P} X)^{-1} (X^{\intercal} \mathbf{P} \varepsilon_Y) = (X^{\intercal} \mathbf{P} X)^{-1} (X^{\intercal} (\mathbf{G}_{\mathcal{V}}^{X \to Y})^{\intercal} (\mathbf{G}_{\mathcal{V}}^{X \to Y} (\mathbf{G}_{\mathcal{V}}^{X \to Y})^{\intercal})^{-1} \mathbf{G}_{\mathcal{V}}^{X \to Y} \varepsilon_Y = 0,$$

which implies that $\hat{\beta} = \beta_{X \to Y}$. $\qquad\square$

## G.2   Proof of Remark 1

*Proof.* Similar to G.1, from procedure "two-stage least squares", we obtained the fitted values $\hat{X}$ from the regression in the first stage: $\hat{X} = (\mathbf{G}_{\mathcal{I}}^{X \to Y})^{\intercal} \hat{\alpha}$. According to OLS regression,

$$\begin{aligned}
\hat{X} &= (\mathbf{G}_{\mathcal{I}}^{X \to Y})^{\intercal} \left[ (\mathbf{G}_{\mathcal{I}}^{X \to Y} (\mathbf{G}_{\mathcal{I}}^{X \to Y})^{\intercal})^{-1} \mathbf{G}_{\mathcal{I}}^{X \to Y} X \right] \\
&= \left[ (\mathbf{G}_{\mathcal{I}}^{X \to Y})^{\intercal} (\mathbf{G}_{\mathcal{I}}^{X \to Y} (\mathbf{G}_{\mathcal{I}}^{X \to Y})^{\intercal})^{-1} \mathbf{G}_{\mathcal{I}}^{X \to Y} \right] X \\
&= \tilde{\mathbf{P}} X.
\end{aligned}$$

Then run the OLS regression for $Y$ on $\hat{X}$ in second stage: $Y = \hat{\beta}\hat{X}$. where:

$$\hat{\beta} = (\hat{X}^\intercal \hat{X})^{-1}(\hat{X}^\intercal Y)$$

$$= ((\tilde{\mathbf{P}}X)^\intercal \tilde{\mathbf{P}}X)^{-1}((\tilde{\mathbf{P}}X)^\intercal Y)$$

$$= (X^\intercal \tilde{\mathbf{P}}X)^{-1}(X^\intercal \tilde{\mathbf{P}}Y)$$

$$= (X^\intercal \tilde{\mathbf{P}}X)^{-1}(X^\intercal \tilde{\mathbf{P}})\left(X\beta_{X\rightarrow Y} + (\mathbf{G}_{\mathcal{I}}^{X\rightarrow Y})^\intercal \boldsymbol{\gamma}_Y + \varepsilon_Y\right)$$

$$= \beta_{X\rightarrow Y} + \underbrace{\left[X^\intercal \tilde{\mathbf{P}}X\right]^{-1} X^\intercal \tilde{\mathbf{P}}(\mathbf{G}_{\mathcal{I}}^{X\rightarrow Y})^\intercal \boldsymbol{\gamma}_Y + \varepsilon_Y)}_{\beta_{bias}}$$

$$= \beta_{X\rightarrow Y} + (X^\intercal \tilde{\mathbf{P}}X)^{-1}(X^\intercal \tilde{\mathbf{P}}(\mathbf{G}_{\mathcal{I}}^{X\rightarrow Y})^\intercal \boldsymbol{\gamma}_Y) + (X^\intercal \tilde{\mathbf{P}}X)^{-1}(X^\intercal \tilde{\mathbf{P}}\varepsilon_Y)$$

Since $\mathbf{G}_{\mathcal{I}}^{X\rightarrow Y}$ is a valid IV set, for each $G_j \in \mathbf{G}_{\mathcal{I}}^{X\rightarrow Y}$, either $\gamma_{Y,j} \neq 0$ (A2), or $cov(G_j, \varepsilon_Y) \neq 0$ (A3), or both $\gamma_{Y,j} \neq 0$ and $cov(G_j, \varepsilon_Y) \neq 0$ (A2 and A3).
Hence, we obtained

$$(X^\intercal \tilde{\mathbf{P}}X)^{-1}(X^\intercal \tilde{\mathbf{P}}(\mathbf{G}_{\mathcal{I}}^{X\rightarrow Y})^\intercal \boldsymbol{\gamma}_Y) \neq 0$$

$$(X^\intercal \tilde{\mathbf{P}}X)^{-1}(X^\intercal \tilde{\mathbf{P}}\varepsilon_Y) = (X^\intercal \tilde{\mathbf{P}}X)^{-1}(X^\intercal (\mathbf{G}_{\mathcal{I}}^{X\rightarrow Y})^\intercal (\mathbf{G}_{\mathcal{I}}^{X\rightarrow Y}(\mathbf{G}_{\mathcal{I}}^{X\rightarrow Y})^\intercal)^{-1}\mathbf{G}_{\mathcal{I}}^{X\rightarrow Y}\varepsilon_Y \neq 0,$$

which implies that $\beta_{bias} \neq 0$. That is to say the estimated causal effect $\hat{\beta} = \beta_{X\rightarrow Y} + \beta_{bias}$ is biased. $\qquad\square$

### G.3 Proof of Proposition 2

*Proof.* We prove it by contradiction, i.e., if $\mathbb{G}$ is a valid IV set for one of the causal relationships, then for each $G_j \in \mathbb{G}$, $\text{corr}(\mathcal{PR}_{(X,Y\,|\,\mathbb{G}\backslash G_j)}, G_j) = 0$. Without loss of generality, we consider the case for $X \rightarrow Y$. The proof with the inverse causal relationship can be derived accordingly.

i) assume that $\mathbb{G}$ is the valid IV set relative to the causal relationship $X \rightarrow Y$, i.e., $\mathbb{G} \subseteq \mathbf{G}_{\mathcal{V}}^{X\rightarrow Y}$. Let $G_j$ be a genetic variant in $\mathbb{G}$. We can see that $\mathbb{G}\backslash G_j$ is also a valid IV set. Because the size of participants $n \rightarrow \infty$, and according to the consistent estimator of TSLS, we have

$$\omega_{\{\mathbb{G}\backslash G_j\}} = \text{TSLS}(X, Y, \mathbb{G}\backslash G_j) = \beta_{X\rightarrow Y}.$$

Based on Eq.(1), we have

$$\mathcal{PR}_{(X,Y\,|\,\mathbb{G}\backslash G_j)} = Y - X\omega_{\{\mathbb{G}\backslash G_j\}} = \mathbf{G}^\intercal \boldsymbol{\gamma}_Y + \varepsilon_Y$$

Because $\mathbb{G}$ is a valid IV set relative to $X \rightarrow Y$, for $G_k \in \mathbf{G}, k \neq j$, either $\gamma_{Y,k} = 0$ (if $G_k \in \mathbb{G}$) or $\text{corr}(G_k, G_j) = 0$ (if $G_k \notin \mathbb{G}$). It is noted that $G_k$ can not be correlated with $G_j$, otherwise, the genetic variant $G_j$ will dissatisfy A2 or A3. Hence, we obtain $\text{corr}(\mathbf{G}^\intercal \boldsymbol{\gamma}_Y + \varepsilon_Y, G_j) = 0$, which implies that $\text{corr}(\mathcal{PR}_{(X,Y\,|\,\mathbb{G}\backslash G_j)}, G_j) = 0$.

ii) assume that $\mathbb{G}$ is the valid IV set relative to $Y \rightarrow X$, i.e., $\mathbb{G} \subseteq \mathbf{G}_{\mathcal{V}}^{Y\rightarrow X}$. Let $G_j$ be a genetic variant in $\mathbb{G}$. We can see that $\mathbb{G}\backslash G_j$ is also a valid IV set relative to $Y \rightarrow X$. It is noted that we here still need to verify Eq.(5). According to the TSLS estimator and Eq.(1), we have

$$\omega_{\{\mathbb{G}\backslash G_j\}} = \text{TSLS}(X, Y, \mathbb{G}\backslash G_j)$$

$$= \left[X^\intercal \tilde{\mathbf{P}}X\right]^{-1} X^\intercal \tilde{\mathbf{P}}Y$$

$$= \left[X^\intercal \tilde{\mathbf{P}}X\right]^{-1} X^\intercal \tilde{\mathbf{P}}(1/\beta_{Y\rightarrow X})(X - \mathbf{G}^\intercal \boldsymbol{\gamma}_X - \varepsilon_X)$$

$$= (1/\beta_{Y\rightarrow X})\left(1 - \left[X^\intercal \tilde{\mathbf{P}}X\right]^{-1} X^\intercal \tilde{\mathbf{P}}(\mathbf{G}^\intercal \boldsymbol{\gamma}_X + \varepsilon_X)\right),$$

$$= \left(1 - \left[X^\intercal \tilde{\mathbf{P}}X\right]^{-1} X^\intercal \tilde{\mathbf{P}}(\mathbf{G}^\intercal \boldsymbol{\gamma}_X + \varepsilon_X)\right)\beta_{Y\rightarrow X}^{-1},$$

where $\tilde{\mathbf{P}} = \{\mathbb{G}\backslash G_j\}\left[\{\mathbb{G}\backslash G_j\}^\intercal \{\mathbb{G}\backslash G_j\}\right]^{-1}\{\mathbb{G}\backslash G_j\}^\intercal$, $X, Y$ and $\varepsilon_X$ are $n \times 1$ vectors, $\mathbb{G}$ is $n \times p$ matrix and $\boldsymbol{\gamma}_X$ is a $g \times 1$ vector.

Hence, we have

$$
\begin{aligned}
\mathcal{PR}_{(X,Y\,|\,\mathbb{G}\backslash G_j)} &= Y - X\omega_{\{\mathbb{G}\backslash G_j\}} \\
&= Y - X\left(1 - \left[X^\mathsf{T}\tilde{\mathbf{P}}X\right]^{-1}X^\mathsf{T}\tilde{\mathbf{P}}(\mathbf{G}^\mathsf{T}\boldsymbol{\gamma}_X + \varepsilon_X)\right)\beta_{Y\to X}^{-1} \\
&= Y - X\beta_{Y\to X}^{-1} + X\left[X^\mathsf{T}\tilde{\mathbf{P}}X\right]^{-1}X^\mathsf{T}\tilde{\mathbf{P}}(\mathbf{G}^\mathsf{T}\boldsymbol{\gamma}_X + \varepsilon_X)\beta_{Y\to X}^{-1} \\
&= -(\mathbf{G}^\mathsf{T}\boldsymbol{\gamma}_X + \varepsilon_X)\beta_{Y\to X}^{-1} + X\left[X^\mathsf{T}\tilde{\mathbf{P}}X\right]^{-1}X^\mathsf{T}\tilde{\mathbf{P}}(\mathbf{G}^\mathsf{T}\boldsymbol{\gamma}_X + \varepsilon_X)\beta_{Y\to X}^{-1} \\
&= \left(-\boldsymbol{I} + X\left[X^\mathsf{T}\tilde{\mathbf{P}}X\right]^{-1}X^\mathsf{T}\tilde{\mathbf{P}}\right)(\mathbf{G}^\mathsf{T}\boldsymbol{\gamma}_X + \varepsilon_X)\,\beta_{Y\to X}^{-1},
\end{aligned}
$$

where $\boldsymbol{I}$ is a $n \times n$ identity matrix.
Because $\mathbb{G}$ is a valid IV set relative to $Y \to X$, for $G_k \in \mathbf{G}, k \neq j$, either $\gamma_{X,k} = 0$ (if $G_k \in \mathbb{G}$) or $\mathrm{corr}(G_k, G_j) = 0$ (if $G_k \notin \mathbb{G}$).

Hence,

$$
\mathrm{corr}(\mathbf{G}^\mathsf{T}\boldsymbol{\gamma}_X + \varepsilon_X), G_j) = 0
$$

Therefore, we obtain

$$
\mathrm{corr}((X/\beta_{Y\to X})\left[X^\mathsf{T}\tilde{\mathbf{P}}X\right]^{-1}X^\mathsf{T}\tilde{\mathbf{P}}(\mathbf{G}^\mathsf{T}\boldsymbol{\gamma}_X + \varepsilon_X), G_j) = 0
$$

which implies that $\mathrm{corr}(\mathcal{PR}_{(X,Y\,|\,\mathbb{G}\backslash G_j)}, G_j) = 0$. $\qquad\square$

### G.4 Proof of Proposition 3

*Proof.* We prove it by contradiction, i.e., if $\mathbb{G}$ is an invalid IV set for one of the causal relationships, then there exists at least one $G_j \in \mathbb{G}$ such that $\mathrm{corr}(\mathcal{PR}_{(X,Y\,|\,\mathbb{G}\backslash G_j)}, G_j) \neq 0$. Without loss of generality, we consider the case for $X \to Y$. The proof with the inverse causal relationship $Y \to X$ can be derived accordingly. There are three cases to be considered. i) Suppose $G_j$ is an invalid IV in $\mathbb{G}$ and $\mathbb{G}\backslash G_j$ is a valid IV set; ii) Suppose $G_j$ is an invalid IV in $\mathbb{G}$ and $\mathbb{G}\backslash G_j$ is an invalid IV set; iii) Suppose $G_j$ is a valid IV in $\mathbb{G}$ and $\mathbb{G}\backslash G_j$ is an invalid IV set.

i) Suppose $G_j$ is invalid and all IVs in $\mathbb{G}\backslash G_j$ are valid. Due to the validity of all IVs in $\mathbb{G}\backslash G_j$, according to the consistent estimator of TSLS, we have,

$$
\omega_{\{\mathbb{G}\backslash G_j\}} = \mathrm{TSLS}(X, Y, \mathbb{G}\backslash G_j) = \beta_{X\to Y}.
$$

In order to consider the case of all violating assumptions for IVs, we extend Eq.(1) to Eq.(6), we have

$$
\mathcal{PR}_{(X,Y\,|\,\mathbb{G}\backslash G_j)} = Y - X\omega_{\{\mathbb{G}\backslash G_j\}} = \mathbf{G}^\mathsf{T}\boldsymbol{\gamma}_Y + \varepsilon_Y = \mathbf{G}^\mathsf{T}\boldsymbol{\gamma}_Y + U\gamma_{Y,U} + \varepsilon_3,
$$

where we make $\varepsilon_Y = U\gamma_{Y,U} + \varepsilon_3$ to tell the influential difference between the latent confounder $U$ and the external noise $\varepsilon_2$ on $Y$. Here $G_j \perp\!\!\!\perp \varepsilon_3$ but $G_j$ may be correlated with $U$ by $\gamma_{U,j}$. Denote by $G_i$ the other IV with $i \neq j$, and we have $G_j \perp\!\!\!\perp G_i$. Hence,

$$
\begin{aligned}
&\mathrm{corr}(\mathcal{PR}_{(X,Y\,|\,\mathbb{G}\backslash G_j)}, G_j) \\
&= \frac{\gamma_{Y,j}cov(G_j, G_j) + \gamma_{Y,U}\gamma_{U,j}cov(G_j, G_j)}{\sqrt{Var(\mathcal{PR}_{(X,Y\,|\,\mathbb{G}\backslash G_j)})}\sqrt{Var(G_j)}} \\
&= \frac{(\gamma_{Y,j} + \gamma_{Y,U}\gamma_{U,j})cov(G_j, G_j)}{\sqrt{Var(\mathcal{PR}_{(X,Y\,|\,\mathbb{G}\backslash G_j)})}\sqrt{Var(G_j)}}.
\end{aligned}
$$

Due to the faithfulness assumption, $\mathrm{corr}(\mathcal{PR}_{(X,Y\,|\,\mathbb{G}\backslash G_j)}, G_j) \neq 0$.

ii) Suppose $G_j$ is invalid and $\mathbb{G}\backslash G_j$ is an invalid IV set, where there are $m$ genetics in $\mathbb{G}\backslash G_j = \{G_1, ..., G_m\}$. Following the procedure of TSLS, we first get the estimate of $X$ with $\mathbb{G}\backslash G_j$, denoted

by $\hat{X}$, and obtain $\omega_{\{\mathbb{G}\setminus G_j\}}$ using $\hat{X}$ and $Y$.

$$\hat{X} = \sum_{k=1}^{m} G_k \hat{\beta}_k = \sum_{k=1}^{m} G_k \Delta[(\gamma_{X,k} + \gamma_{Y,k}\beta_{Y\to X}) + (\gamma_{X,U} + \gamma_{Y,U}\beta_{Y\to X})\gamma_{U,k}].$$

$$
\begin{aligned}
\omega_{\{\mathbb{G}\setminus G_j\}} &= \frac{\sum_{k=1}^{m}[\hat{\beta}_k(\gamma_{Y,k} + \gamma_{X,k}\beta_{X\to Y}) + \hat{\beta}_k\gamma_{U,k}(\gamma_{Y,U} + \gamma_{X,U}\beta_{X\to Y})]\mathbf{C}_{kk}\Delta}{\sum_{k=1}^{m}(\hat{\beta}_k)^2} \\
&= \frac{\sum_{k=1}^{m}[\bar{\beta}_k(\gamma_{Y,k} + \gamma_{X,k}\beta_{X\to Y}) + \bar{\beta}_k\gamma_{U,k}(\gamma_{Y,U} + \gamma_{X,U}\beta_{X\to Y})]\mathbf{C}_{kk}}{\sum_{k=1}^{m}(\bar{\beta}_m)^2}, \\
&= \frac{\sum_{k=1}^{m}\bar{\beta}_k[(\gamma_{Y,k} + \gamma_{X,k}\beta_{X\to Y}) + \gamma_{U,k}(\gamma_{Y,U} + \gamma_{X,U}\beta_{X\to Y})]\mathbf{C}_{kk}}{\sum_{k=1}^{m}(\bar{\beta}_k)^2} \\
&= \frac{\sum_{k=1}^{m}(A_{X,k} + A_{X,U}\gamma_{U,k})(A_{Y,k} + A_{Y,U}\gamma_{U,k})\mathbf{C}_{kk}}{\sum_{k=1}^{m}(\bar{\beta}_k)^2},
\end{aligned}
$$

where $\bar{\beta}_k = \hat{\beta}_k/\Delta = (\gamma_{X,k} + \gamma_{Y,k}\beta_{Y\to X}) + \gamma_{U,k}(\gamma_{X,U} + \gamma_{Y,U}\beta_{Y\to X}) = A_{X,k} + A_{X,U}\gamma_{U,k}$, $A_{X,k} = \gamma_{X,k} + \gamma_{Y,k}\beta_{Y\to X}$ and $A_{X,U} = \gamma_{X,U} + \gamma_{Y,U}\beta_{Y\to X}$. $A_{Y,k} = \gamma_{Y,k} + \gamma_{X,k}\beta_{X\to Y}$ and $A_{Y,U} = \gamma_{Y,U} + \gamma_{X,U}\beta_{X\to Y}$ and $\mathbf{C}_{kk} = cov(G_k, G_k)$. Note that such notations $A.$ enjoy some great properties shown as below,

$$
\begin{aligned}
A_{X,k} - A_{Y,k}\beta_{Y\to X} &= \gamma_{X,k}(1 - \beta_{X\to Y}\beta_{Y\to X}), \\
A_{X,k}\beta_{X\to Y} - A_{Y,k} &= \gamma_{Y,k}(\beta_{X\to Y}\beta_{Y\to X} - 1), \\
A_{X,1}A_{Y,2} - A_{X,2}A_{Y,1} &= (1 - \beta_{X\to Y}\beta_{Y\to X})(\gamma_{X,1}\gamma_{Y,2} - \gamma_{X,2}\gamma_{Y,1}).
\end{aligned}
\tag{19}
$$

Hence,

$$
\begin{aligned}
&\text{corr}(\mathcal{PR}_{(X,Y\,|\,\mathbb{G}\setminus G_j)}, G_j) \\
&= \frac{cov(G_j, G_j)}{\sqrt{Var(\mathcal{PR}_{(X,Y\,|\,\mathbb{G}\setminus G_j)})}\sqrt{Var(G_j)}}\{(\gamma_{Y,j} + \gamma_{X,j}\beta_{X\to Y})\Delta - (\gamma_{X,j} + \gamma_{Y,j}\beta_{Y\to X})\Delta\omega_{\{\mathbb{G}\setminus G_j\}} \\
&\quad + \gamma_{U,j}(\gamma_{Y,U} + \gamma_{X,U}\beta_{X\to Y})\Delta - \gamma_{U,j}(\gamma_{X,U} + \gamma_{Y,U}\beta_{Y\to X})\Delta\omega_{\{\mathbb{G}\setminus G_j\}}\} \\
&= \frac{cov(G_j, G_j)\Delta}{\sqrt{Var(\mathcal{PR}_{(X,Y\,|\,\mathbb{G}\setminus G_j)})}\sqrt{Var(G_j)}}\{[\gamma_{Y,j}(1 - \beta_{Y\to X}\omega_{\{\mathbb{G}\setminus G_j\}}) + \gamma_{X,j}(\beta_{X\to Y} - \omega_{\{\mathbb{G}\setminus G_j\}})] \\
&\quad + \gamma_{U,j}[\gamma_{Y,U}(1 - \beta_{Y\to X}\omega_{\{\mathbb{G}\setminus G_j\}}) + \gamma_{X,U}(\beta_{X\to Y} - \omega_{\{\mathbb{G}\setminus G_j\}})]\} \\
&= \delta\Delta[(1 - \beta_{Y\to X}\omega_{\{\mathbb{G}\setminus G_j\}})(\gamma_{Y,j} + \gamma_{Y,U}\gamma_{U,j}) + (\beta_{X\to Y} - \omega_{\{\mathbb{G}\setminus G_j\}})(\gamma_{X,j} + \gamma_{X,U}\gamma_{U,j})],
\end{aligned}
$$

where $\delta = \frac{\sqrt{Var(G_j)}}{\sqrt{Var(\mathcal{PR}_{(X,Y\,|\,\mathbb{G}\setminus G_j)})}} \neq 0$. Employing the properties in Eqs.(19), we get

$$
\begin{aligned}
&1 - \beta_{Y\to X}\omega_{\{\mathbb{G}\setminus G_j\}} \\
&= \frac{\sum_{k=1}^{m}(A_{X,k} + A_{X,U}\gamma_{U,k})^2\mathbf{C}_{kk} - \beta_{Y\to X}(A_{X,k} + A_{X,U}\gamma_{U,k})(A_{Y,k} + A_{Y,U}\gamma_{U,k})\mathbf{C}_{kk}}{\sum_{k=1}^{m}(\bar{\beta}_k)^2\mathbf{C}_{kk}} \\
&= \frac{(1 - \beta_{X\to Y}\beta_{Y\to X})\sum_{k=1}^{m}(A_{X,k} + A_{X,U}\gamma_{U,k})(\gamma_{X,k} + \gamma_{X,U}\gamma_{U,k})\mathbf{C}_{kk}}{\sum_{k=1}^{m}(\bar{\beta}_k)^2\mathbf{C}_{kk}},
\end{aligned}
$$

and

$$
\begin{aligned}
&\beta_{X\to Y} - \omega_{\{\mathbb{G}\setminus G_j\}} \\
&= \frac{\beta_{X\to Y}\sum_{k=1}^{m}(A_{X,k} + A_{X,U}\gamma_{U,k})^2\mathbf{C}_{kk} - \sum_{k=1}^{m}(A_{X,k} + A_{X,U}\gamma_{U,k})(A_{Y,k} + A_{Y,U}\gamma_{U,k})\mathbf{C}_{kk}}{\sum_{k=1}^{m}(\bar{\beta}_k)^2\mathbf{C}_{kk}} \\
&= \frac{(\beta_{X\to Y}\beta_{Y\to X} - 1)\sum_{k=1}^{m}(A_{X,k} + A_{X,U}\gamma_{U,k})(\gamma_{Y,k} + \gamma_{Y,U}\gamma_{U,k})\mathbf{C}_{kk}}{\sum_{k=1}^{m}(\bar{\beta}_k)^2\mathbf{C}_{kk}}.
\end{aligned}
$$

Thus,

$$
\begin{aligned}
&\mathrm{corr}(\mathcal{PR}_{(X,Y\,|\,\mathbb{G}\backslash G_j)}, G_j)\\
&= \delta\{\frac{\sum_{k=1}^m \mathbf{C}_{kk}(A_{X,k}+A_{X,U}\gamma_{U,k})(\gamma_{X,k}+\gamma_{X,U}\gamma_{U,k})}{\sum_{k=1}^m(\bar{\beta}_k)^2\mathbf{C}_{kk}}(\gamma_{Y,j}+\gamma_{Y,U}\gamma_{U,j})\\
&\quad - \frac{\sum_{k=1}^m \mathbf{C}_{kk}(A_{X,k}+A_{X,U}\gamma_{U,k})(\gamma_{Y,k}+\gamma_{Y,U}\gamma_{U,k})}{\sum_{k=1}^m(\bar{\beta}_k)^2\mathbf{C}_{kk}}(\gamma_{X,j}+\gamma_{X,U}\gamma_{U,j})\}\\
&= \delta\frac{\sum_{k=1}^m \bar{\beta}_k\mathbf{C}_{kk}[(\gamma_{X,k}+\gamma_{X,U}\gamma_{U,k})(\gamma_{Y,j}+\gamma_{Y,U}\gamma_{U,j})-(\gamma_{Y,k}+\gamma_{Y,U}\gamma_{U,k})(\gamma_{X,j}+\gamma_{X,U}\gamma_{U,j})]}{\sum_{k=1}^m(\bar{\beta}_k)^2\mathbf{C}_{kk}}\\
&= \delta\frac{\sum_{k=1}^m \bar{\beta}_k\mathbf{C}_{kk}[(\gamma_{X,k}\gamma_{Y,j}-\gamma_{Y,k}\gamma_{X,j})+\gamma_{Y,U}(\gamma_{X,k}\gamma_{U,j}-\gamma_{U,k}\gamma_{X,j})+\gamma_{X,U}(\gamma_{U,k}\gamma_{Y,j}-\gamma_{Y,k}\gamma_{U,j})]}{\sum_{k=1}^m(\bar{\beta}_k)^2\mathbf{C}_{kk}}\neq 0.
\end{aligned}
$$

Because of the faithfulness assumption, $\bar{\beta}_k \neq 0$. Due to Assumption 2, for $G_k \in \mathbb{G}\backslash G_j$ and the confounder $U$, $\sum_{k=1}^m \bar{\beta}_k\mathbf{C}_{kk}[(\gamma_{X,k}\gamma_{Y,j}-\gamma_{Y,k}\gamma_{X,j})+\gamma_{Y,U}(\gamma_{X,k}\gamma_{U,j}-\gamma_{U,k}\gamma_{X,j})+\gamma_{X,U}(\gamma_{U,k}\gamma_{Y,j}-\gamma_{Y,k}\gamma_{U,j})] \neq 0$.

iii) Suppose $G_j$ is a valid IV in $\mathbb{G}$ and $\mathbb{G}\backslash G_j$ is an invalid IV set. Similar to ii), we get $\omega_{\{\mathbb{G}\backslash G_j\}}$:

$$
\omega_{\{\mathbb{G}\backslash G_j\}} = \frac{\sum_{k=1}^m(A_{X,k}+A_{X,U}\gamma_{U,k})(A_{Y,k}+A_{Y,U}\gamma_{U,k})\mathbf{C}_{kk}}{\sum_{k=1}^m(\bar{\beta}_k)^2\mathbf{C}_{kk}}.
$$

Since the valid IV $G_j$ is uncorrelated with neither other invalid IVs nor the confounder, we then derive,

$$
\begin{aligned}
&\mathrm{corr}(\mathcal{PR}_{(X,Y\,|\,\mathbb{G}\backslash G_j)}, G_j)\\
&= \frac{cov(G_j,G_j)\Delta[(\gamma_{Y,j}+\gamma_{X,j}\beta_{X\to Y})-(\gamma_{X,j}+\gamma_{Y,j}\beta_{Y\to X})\Delta\omega_{\{\mathbb{G}\backslash G_j\}}]}{\sqrt{Var(\mathcal{PR}_{(X,Y\,|\,\mathbb{G}\backslash G_j)})}\sqrt{Var(G_j)}}\\
&= \delta\Delta\gamma_{X,j}(\beta_{X\to Y}-\omega_{\{\mathbb{G}\backslash G_j\}}).
\end{aligned}
$$

where:

$$
\begin{aligned}
&\beta_{X\to Y}-\omega_{\{\mathbb{G}\backslash G_j\}}\\
&= \frac{\beta_{X\to Y}\sum_{k=1}^m \mathbf{C}_{kk}[(A_{X,k}+A_{X,U}\gamma_{U,k})^2-(A_{X,k}+A_{X,U}\gamma_{U,k})(A_{Y,k}+A_{Y,U}\gamma_{U,k})]}{\sum_{k=1}^m(\bar{\beta}_k)^2\mathbf{C}_{kk}}\\
&= \frac{(\beta_{X\to Y}\beta_{Y\to X}-1)\sum_{k=1}^m \mathbf{C}_{kk}(A_{X,k}+A_{X,U}\gamma_{U,k})(\gamma_{Y,k}+\gamma_{Y,U}\gamma_{U,k})}{\sum_{k=1}^m(\bar{\beta}_k)^2\mathbf{C}_{kk}}.
\end{aligned}
$$

Thus, we get

$$
\begin{aligned}
&\mathrm{corr}(\mathcal{PR}_{(X,Y\,|\,\mathbb{G}\backslash G_j)}, G_j) = \delta\Delta\gamma_{X,j}(\beta_{X\to Y}-\omega_{\{\mathbb{G}\backslash G_j\}})\\
&= -\delta\frac{\sum_{k=1}^m \mathbf{C}_{kk}(A_{X,k}+A_{X,U}\gamma_{U,k})(\gamma_{Y,k}+\gamma_{Y,U}\gamma_{U,k})\gamma_{X,j}}{\sum_{k=1}^m(\bar{\beta}_k)^2\mathbf{C}_{kk}}\\
&= -\delta\frac{\sum_{k=1}^m \bar{\beta}_k\mathbf{C}_{kk}(\gamma_{Y,k}+\gamma_{Y,U}\gamma_{U,k})\gamma_{X,j}}{\sum_{k=1}^m(\bar{\beta}_k)^2\mathbf{C}_{kk}}.
\end{aligned}
$$

Since $G_j$ is a valid IV ($\gamma_{Y,j}=\gamma_{U,j}=0$), and from Assumption 2, we have $\sum_{k=1}^m \bar{\beta}_k\mathbf{C}_{kk}(\gamma_{Y,k}+\gamma_{Y,U}\gamma_{U,k})\gamma_{X,j} \neq 0$. Proposition 3 is proved. □

### G.5 Proof of Proposition 4

*Proof.* We prove this proposition by contradiction, i.e., i) if $\mathbb{G}$ is the valid IV set relative to the causal relationship $Y \to X$, then $\exists\, G_j \in \mathbb{G}$ satisfies $|\frac{\mathrm{corr}(G_j,Y)}{\mathrm{corr}(G_j,X)}| \geq 1$; and ii) if $\mathbb{G}$ is the valid IV set relative to the causal relationship $X \to Y$, then $\exists\, G_j \in \mathbb{G}$ satisfies $|\frac{\mathrm{corr}(G_j,Y)}{\mathrm{corr}(G_j,X)}| < 1$.

i) Suppose $\mathbb{G}$ is the valid IV set relative to the causal relationship $Y \to X$. According to Assumption 3, we have $|\beta_{Y \to X}| \leq \sqrt{\frac{\text{Var}(X)}{\text{Var}(Y)}}$. Hence,

$$
\left| \frac{\text{corr}(G_j, Y)}{\text{corr}(G_j, X)} \right| = \left| \frac{\text{Cov}(G_j, Y)}{\sqrt{\text{Var}(G_j)\text{Var}(Y)}} \frac{\sqrt{\text{Var}(G_j)\text{Var}(X)}}{\text{Cov}(G_j, X)} \right|
$$

$$
= \left| \frac{\gamma_{Y,j}\Delta}{\sqrt{\text{Var}(Y)}} \frac{\sqrt{\text{Var}(X)}}{\gamma_{Y,j}\Delta\beta_{Y \to X}} \right|
$$

$$
= \left| \frac{1}{\beta_{Y \to X}} \sqrt{\frac{\text{Var}(X)}{\text{Var}(Y)}} \right| \geq 1,
$$

which proves case i).

ii) Suppose $\mathbb{G}$ is the valid IV set relative to the causal relationship $X \to Y$. According to Assumption 3, we have $|\beta_{X \to Y}| < \sqrt{\frac{\text{Var}(Y)}{\text{Var}(X)}}$. Hence,

$$
\left| \frac{\text{corr}(G_j, Y)}{\text{corr}(G_j, X)} \right| = \left| \frac{\text{Cov}(G_j, Y)}{\sqrt{\text{Var}(G_j)\text{Var}(Y)}} \frac{\sqrt{\text{Var}(G_j)\text{Var}(X)}}{\text{Cov}(G_j, X)} \right|
$$

$$
= \left| \frac{\gamma_{X,j}\Delta\beta_{X \to Y}}{\sqrt{\text{Var}(Y)}} \frac{\sqrt{\text{Var}(X)}}{\gamma_{X,j}\Delta} \right|
$$

$$
= \left| \beta_{X \to Y} \sqrt{\frac{\text{Var}(X)}{\text{Var}(Y)}} \right| < 1,
$$

which proves case ii) and concludes the proposition. $\square$

### G.6 Proof of Theorem 1

*Proof.* Firstly, given Assumptions 1 and 2, in light of Propositions 2 and 3, we can identify all sets of valid instrumental variables (IVs). Next, given Assumption 3, by Proposition 4, we can further identify what the causal relationships corresponding to these IV sets. It is important to note that if all the IV sets correspond to a single causal relationship, such as $X \to Y$, we can conclude that the model is a one-directional MR model. Finally, with the corresponding IV set identified for each causal influence, we can employ the classical IV estimator, namely the two-stage least squares (TSLS) estimator, to accurately estimate the causal effects.

Based on the above analysis, we can conclude that the bi-directional MR model is fully identifiable.

$\square$

### G.7 Proof of Theorem 2

*Proof.* The correctness of PReBiM originates from the following observations:

- Firstly, in Step 1 (Algorithm 4), by Propositions 2 and 3, valid IV sets in $\mathbf{G}$ have been exactly discovered in $\mathcal{V}$ (Lines $2 \sim 25$ of Algorithm 4).

- Secondly, in Step 2 (Algorithm 5), by the condition of Proposition 4, the direction of causal influence can be determined (Line $2 \sim 8$ of Algorithm 5).

- Lastly, by Eq.(1) of Proposition 1, unbiased causal effects can be estimated exactly given the identified IV sets (Lines $9 \sim 10$ of Algorithm 5).

Based on the above analysis, we can identify all valid IV sets and obtain the causal effects correctly.

$\square$

## H   More Experimental Results

### H.1   Evaluating the Performance on Bi-directional MR Model with Dependent Valid IVs

Table 3 summarizes the CSR and MSE metrics' results across six methods within a bi-directional MR model with varying sample sizes. As expected, our proposed method exhibits superior performance, even in scenarios where the included valid instrumental variables (IVs) are correlated. This highlights its capability to accurately identify effective IVs and provide consistent causal effect estimates solely from observational data, regardless of whether the relevant IVs are correlated. Importantly, this effectiveness persists without prior knowledge of the causal direction between $X$ and $Y$. Conversely, the performance of the other three methods remains suboptimal in such contexts, indicating their limited applicability in cases with correlated valid IVs.

Table 3: Performance comparison of NAIVE, MR-Egger, sisVIVE, IV-TETRAD, TSHT, and PReBiM in estimating bi-directional MR models across various sample sizes and three scenarios.

| | | $\mathcal{S}(2,2,6)$ | | | | $\mathcal{S}(3,3,8)$ | | | | $\mathcal{S}(4,4,10)$ | | | |
| | | $X \to Y$ | | $Y \to X$ | | $X \to Y$ | | $Y \to X$ | | $X \to Y$ | | $Y \to X$ | |
| Size | Algorithm | CSR↑ | MSE↓ | CSR↑ | MSE↓ | CSR↑ | MSE↓ | CSR↑ | MSE↓ | CSR↑ | MSE↓ | CSR↑ | MSE↓ |
|---|---|---|---|---|---|---|---|---|---|---|---|---|---|
| 2k | NAIVE | - | 0.409 | - | 0.341 | - | 0.350 | - | 0.372 | - | 0.409 | - | 0.314 |
| | MR-Egger | - | 1.198 | - | 1.320 | - | 1.033 | - | 1.322 | - | 1.087 | - | 1.391 |
| | sisVIVE | 0.31 | 0.478 | 0.29 | 0.395 | 0.34 | 0.408 | 0.34 | 0.388 | 0.35 | 0.450 | 0.36 | 0.328 |
| | IV-TETRAD | 0.40 | 1.224 | 0.42 | 0.900 | 0.52 | 0.830 | 0.29 | 0.903 | 0.44 | 1.033 | 0.32 | 0.779 |
| | TSHT | 0.09 | 0.657 | 0.04 | 0.636 | 0.10 | 0.696 | 0.08 | 0.699 | 0.12 | 0.804 | 0.10 | 0.708 |
| | PReBiM | **0.84** | **0.014** | **0.79** | **0.167** | **0.89** | **0.058** | **0.88** | **0.097** | **0.84** | **0.053** | **0.83** | **0.079** |
| 5k | NAIVE | - | 0.399 | - | 0.326 | - | 0.371 | - | 0.362 | - | 0.391 | - | 0.333 |
| | MR-Egger | - | 1.000 | - | 1.204 | - | 1.171 | - | 1.181 | - | 0.971 | - | 1.358 |
| | sisVIVE | 0.30 | 0.484 | 0.31 | 0.426 | 0.35 | 0.452 | 0.34 | 0.381 | 0.38 | 0.375 | 0.37 | 0.327 |
| | IV-TETRAD | 0.50 | 1.042 | 0.40 | 0.700 | 0.49 | 1.041 | 0.38 | 0.778 | 0.47 | 0.977 | 0.36 | 0.778 |
| | TSHT | 0.06 | 0.701 | 0.05 | 0.531 | 0.09 | 0.689 | 0.06 | 0.660 | 0.11 | 0.653 | 0.09 | 0.589 |
| | PReBiM | **0.90** | **0.113** | **0.84** | **0.081** | **0.91** | **0.050** | **0.90** | **0.061** | **0.88** | **0.048** | **0.89** | **0.061** |
| 10k | NAIVE | - | 0.378 | - | 0.358 | - | 0.408 | - | 0.323 | - | 0.401 | - | 0.339 |
| | MR-Egger | - | 1.052 | - | 1.307 | - | 1.017 | - | 1.313 | - | 1.185 | - | 1.303 |
| | sisVIVE | 0.33 | 0.465 | 0.34 | 0.427 | 0.35 | 0.448 | 0.34 | 0.358 | 0.38 | 0.431 | 0.41 | 0.299 |
| | IV-TETRAD | 0.54 | 0.995 | 0.39 | 0.839 | 0.52 | 1.092 | 0.41 | 0.637 | 0.51 | 0.963 | 0.38 | 0.553 |
| | TSHT | 0.04 | 0.641 | 0.03 | 0.543 | 0.04 | 0.734 | 0.04 | 0.449 | 0.09 | 0.654 | 0.07 | 0.567 |
| | PReBiM | **0.90** | **0.058** | **0.87** | **0.089** | **0.94** | **0.018** | **0.93** | **0.026** | **0.93** | **0.022** | **0.92** | **0.056** |

Note: This scenario represents a bi-directional model with correlated valid IVs. The symbol "−" means that methods do not output information. "↑" means a higher value is better, and vice versa.

### H.2   Evaluating the Performance on One-directional MR Model with Dependent Valid IVs

In this section, we assess the efficacy of four methods within a one-directional Mendelian Randomization (MR) framework. For a fair comparison, we operate under the assumption that the causal pathway from $X$ to $Y$ is established for all methods. Figure 7 displays the CSR and MSE metrics for these methods, considering diverse sample sizes and numbers of valid instrumental variables (IVs). As expected, our approach maintains strong performance, even amidst correlated valid IVs. The outcomes are comparable with those of the TSHT algorithm, with our method notably surpassing both the sisVIVE and IV-TETRAD algorithms under various conditions.

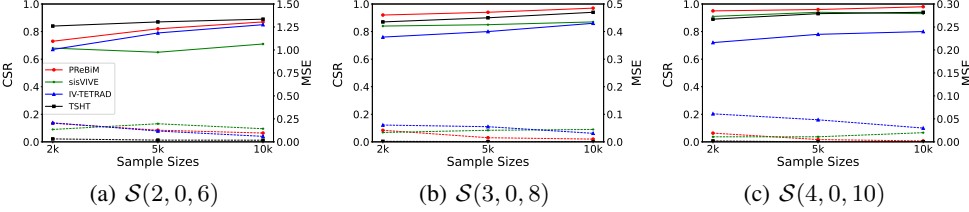

(a) $\mathcal{S}(2,0,6)$      (b) $\mathcal{S}(3,0,8)$      (c) $\mathcal{S}(4,0,10)$

Figure 7: Performance comparison of sisVIVE, IV-TETRAD, TSHT, and PReBiM in estimating one-directional MR models incorporating valid IVs across various sample sizes and three scenarios.

### H.3   Evaluating the Performance on Bi-directional MR Model with Small Sample Sizes

In this section, we will evaluate the performance of these six methods in a bi-directional Mendelian randomization framework with small samples. Table 4 displays the CSR and MSE metrics for

these methods, considering three scenarios with a sample size of 200. We find that though the performances of all four methods are not satisfactory, ours still outperforms other baselines. It also reflects an open problem of our work with small sample sizes. We leave it as our future in-depth direction.

Table 4: Performance comparison of NAIVE, MR-Egger, sisVIVE, IV-TETRAD, TSHT, and PReBiM in estimating bi-directional MR models across small sample size (200) and three scenarios.

| | | $\mathcal{S}(2,2,6)$ | | | | $\mathcal{S}(3,3,8)$ | | | | $\mathcal{S}(4,4,10)$ | | | |
| | | $X \rightarrow Y$ | | $Y \rightarrow X$ | | $X \rightarrow Y$ | | $Y \rightarrow X$ | | $X \rightarrow Y$ | | $Y \rightarrow X$ | |
| Size | Algorithm | CSR↑ | MSE↓ | CSR↑ | MSE↓ | CSR↑ | MSE↓ | CSR↑ | MSE↓ | CSR↑ | MSE↓ | CSR↑ | MSE↓ |
|---|---|---|---|---|---|---|---|---|---|---|---|---|---|
| | NAIVE | - | 0.360 | - | 0.330 | - | 0.343 | - | 0.331 | - | 0.320 | - | 0.336 |
| | MR-Egger | - | 0.941 | - | 0.657 | - | 0.579 | - | 0.594 | - | 0.433 | - | 0.535 |
| 200 | sisVIVE | 0.21 | 0.389 | 0.24 | 0.383 | 0.29 | 0.331 | 0.28 | 0.336 | 0.34 | 0.318 | 0.30 | 0.329 |
| | IV-TETRAD | 0.30 | 1.057 | 0.26 | 0.797 | 0.26 | 0.811 | 0.20 | 0.886 | 0.15 | 0.723 | 0.12 | 0.787 |
| | TSHT | 0.17 | 0.677 | 0.16 | 0.642 | 0.22 | 0.504 | 0.24 | 0.534 | 0.26 | 0.493 | 0.23 | 0.452 |
| | PReBiM | **0.54** | **0.339** | **0.55** | **0.350** | **0.55** | **0.313** | **0.57** | **0.281** | **0.51** | **0.270** | **0.52** | **0.288** |

Note: The symbol "−" means that methods do not output this information. "↑" means a higher value is better, and vice versa.

## H.4 Evaluating the Performance on Bi-directional MR Model with Different Numbers of Valid IVs

Table 5 summarizes the CSR and MSE metrics' results across six methods within a bi-directional MR model with different numbers of valid IVs. Specifically, $\mathcal{S}(2,3,7)$ indicates that in the bi-directional model, there are two valid instrumental variables estimated for the direction $X \rightarrow Y$, and three valid instrumental variables estimated for the direction $Y \rightarrow X$, with the remaining two being invalid instrumental variables. The same interpretation applies to $\mathcal{S}(2,4,8)$ and $\mathcal{S}(3,5,10)$. It can be seen that whether the number of valid IVs in these two directions is the same or not does not affect the performance of our algorithm theoretically. The experimental results also empirically verify it. Overall, In cases where the number of IVs in both the $X \rightarrow Y$ and $Y \rightarrow X$ directions are different, our PReBiM algorithm outperforms other baselines in terms of CSR and MSE.

Table 5: Performance comparison of NAIVE, MR-Egger, sisVIVE, IV-TETRAD, TSHT, and PReBiM in estimating bi-directional MR models across various sample sizes and three scenarios.

| | | $\mathcal{S}(2,3,7)$ | | | | $\mathcal{S}(2,4,8)$ | | | | $\mathcal{S}(3,5,10)$ | | | |
| | | $X \rightarrow Y$ | | $Y \rightarrow X$ | | $X \rightarrow Y$ | | $Y \rightarrow X$ | | $X \rightarrow Y$ | | $Y \rightarrow X$ | |
| Size | Algorithm | CSR↑ | MSE↓ | CSR↑ | MSE↓ | CSR↑ | MSE↓ | CSR↑ | MSE↓ | CSR↑ | MSE↓ | CSR↑ | MSE↓ |
|---|---|---|---|---|---|---|---|---|---|---|---|---|---|
| | NAIVE | - | 0.433 | - | 0.267 | - | 0.464 | - | 0.227 | - | 0.431 | - | 0.237 |
| | MR-Egger | - | 0.702 | - | 0.646 | - | 0.560 | - | 0.610 | - | 0.465 | - | 0.476 |
| 2k | sisVIVE | 0.16 | 0.746 | 0.52 | 0.170 | 0.11 | 0.958 | 0.68 | 0.093 | 0.14 | 0.907 | 0.65 | 0.099 |
| | IV-TETRAD | 0.37 | 1.490 | 0.55 | 0.368 | 0.24 | 1.886 | 0.71 | 0.188 | 0.35 | 1.474 | 0.56 | 0.150 |
| | TSHT | 0.00 | 2.226 | 0.72 | 0.066 | 0.00 | 2.467 | 0.85 | 0.004 | 0.00 | 2.453 | 0.84 | 0.003 |
| | PReBiM | **0.81** | **0.062** | **0.89** | **0.036** | **0.78** | **0.063** | **0.89** | **0.031** | **0.80** | **0.053** | **0.87** | **0.038** |
| | NAIVE | - | 0.394 | - | 0.307 | - | 0.479 | - | 0.228 | - | 0.457 | - | 0.228 |
| | MR-Egger | - | 0.509 | - | 0.691 | - | 0.593 | - | 0.546 | - | 0.540 | - | 0.486 |
| 5k | sisVIVE | 0.20 | 0.611 | 0.49 | 0.274 | 0.09 | 1.085 | 0.72 | 0.113 | 0.13 | 0.917 | 0.66 | 0.089 |
| | IV-TETRAD | 0.37 | 1.479 | 0.56 | 0.312 | 0.25 | 1.869 | 0.69 | 0.165 | 0.30 | 1.641 | 0.63 | 0.164 |
| | TSHT | 0.00 | 2.296 | 0.80 | 0.027 | 0.00 | 2.476 | 0.90 | 0.001 | 0.00 | 2.477 | 0.90 | 0.001 |
| | PReBiM | **0.86** | **0.067** | **0.93** | **0.008** | **0.83** | **0.035** | **0.93** | **0.023** | **0.86** | **0.021** | **0.91** | **0.003** |
| | NAIVE | - | 0.434 | - | 0.273 | - | 0.463 | - | 0.234 | - | 0.434 | - | 0.226 |
| | MR-Egger | - | 0.646 | - | 0.593 | - | 0.550 | - | 0.560 | - | 0.525 | - | 0.443 |
| 10k | sisVIVE | 0.17 | 0.807 | 0.50 | 0.230 | 0.10 | 1.083 | 0.71 | 0.088 | 0.15 | 0.932 | 0.77 | 0.079 |
| | IV-TETRAD | 0.38 | 1.495 | 0.59 | 0.322 | 0.26 | 1.753 | 0.69 | 0.244 | 0.34 | 1.429 | 0.62 | 0.204 |
| | TSHT | 0.00 | 2.370 | 0.85 | 0.022 | 0.00 | 2.494 | 0.92 | 0.001 | 0.00 | 2.486 | 0.93 | 0.000 |
| | PReBiM | **0.90** | **0.022** | **0.97** | **0.005** | **0.91** | **0.013** | **0.95** | **0.009** | **0.92** | **0.038** | **0.93** | **0.008** |

Note: The symbol "−" means that methods do not output this information. "↑" means a higher value is better, and vice versa.

## H.5 Evaluating the Performance on Bi-directional MR Model with Large-Scale Datasets

In this section, we evaluate the performance of these methods in a bi-directional MR framework with large-scale datasets. Table 6 show results of the settings $\mathcal{S}(10,10,30)$ and $\mathcal{S}(15,15,40)$. We observe that the overall performances of all methods decrease with large-scale datasets, whereas our method still performs more superior than other baselines.

Table 6: Performance comparison of NAIVE, MR-Egger, sisVIVE, IV-TETRAD, TSHT, and PRe-BiM in estimating bi-directional MR models with large-scale datasets.

| Size | Algorithm | $\mathcal{S}(10,10,30)$ | | | | $\mathcal{S}(15,15,40)$ | | | |
| | | $X \to Y$ | | $Y \to X$ | | $X \to Y$ | | $Y \to X$ | |
| | | CSR↑ | MSE↓ | CSR↑ | MSE↓ | CSR↑ | MSE↓ | CSR↑ | MSE↓ |
|---|---|---|---|---|---|---|---|---|---|
| | NAIVE | - | 0.298 | - | 0.292 | - | 0.288 | - | 0.299 |
| | MR-Egger | - | 0.337 | - | 0.316 | - | 0.276 | - | 0.285 |
| 2k | sisVIVE | 0.15 | 0.275 | 0.15 | 0.246 | 0.24 | 0.188 | 0.22 | 0.223 |
| | IV-TETRAD | 0.22 | 0.802 | 0.22 | 0.647 | 0.11 | 0.666 | 0.07 | 0.700 |
| | TSHT | 0.18 | 1.029 | 0.28 | 0.874 | 0.27 | 1.012 | 0.28 | 0.995 |
| | PReBiM | **0.59** | **0.244** | **0.57** | **0.173** | **0.67** | **0.152** | **0.71** | **0.157** |
| | NAIVE | - | 0.297 | - | 0.299 | - | 0.301 | - | 0.285 |
| | MR-Egger | - | 2.307 | - | 0.323 | - | 0.293 | - | 0.278 |
| 5k | sisVIVE | 0.15 | 0.247 | 0.14 | 0.281 | 0.23 | 0.236 | 0.23 | 0.200 |
| | IV-TETRAD | 0.49 | 0.635 | 0.48 | 0.495 | 0.38 | 0.572 | 0.36 | 0.546 |
| | TSHT | 0.20 | 1.020 | 0.21 | 1.083 | 0.22 | 0.967 | 0.19 | 1.042 |
| | PReBiM | **0.72** | **0.079** | **0.73** | **0.102** | **0.81** | **0.094** | **0.77** | **0.075** |
| | NAIVE | - | 0.301 | - | 0.284 | - | 0.307 | - | 0.279 |
| | MR-Egger | - | 0.338 | - | 0.325 | - | 0.280 | - | 0.272 |
| 10k | sisVIVE | 0.16 | 0.271 | 0.17 | 0.264 | 0.24 | 0.216 | 0.25 | 0.196 |
| | IV-TETRAD | 0.58 | 0.443 | 0.62 | 0.390 | 0.53 | 0.535 | 0.53 | 0.432 |
| | TSHT | 0.19 | 0.957 | 0.20 | 0.863 | 0.22 | 0.886 | 0.18 | 1.024 |
| | PReBiM | **0.80** | **0.062** | **0.80** | **0.062** | **0.85** | **0.061** | **0.84** | **0.075** |

Note: The symbol "$-$" means that methods do not output this information. "↑" means a higher value is better, and vice versa.

## H.6 Evaluating the Performance on Bi-directional MR Model with All IVs Being Valid

In this section, we evaluate the performance of these methods in a bi-directional MR framework with all IVs being valid. That is, only valid IVs are included in the model, either to estimate the direction $X \to Y$ or the direction $Y \to X$. The results in Table 7 below show the settings $\mathcal{S}(2,2,4)$. To ensure the thoroughness of the experiment, we've added another setting, $\mathcal{S}(3,3,6)$. As expected, our method gives the best results across all sample sizes and directions. As the sample size increases, the metric CSR approaches 1, indicating that our method can identify valid IVs and also validating our theoretical guarantees.

Table 7: Performance comparison of NAIVE, MR-Egger, sisVIVE, IV-TETRAD, TSHT, and PRe-BiM in estimating bi-directional MR models across various sample sizes and three scenarios.

| Size | Algorithm | $\mathcal{S}(2,2,4)$ $X \rightarrow Y$ CSR↑ | MSE↓ | $Y \rightarrow X$ CSR↑ | MSE↓ | $\mathcal{S}(3,3,6)$ $X \rightarrow Y$ CSR↑ | MSE↓ | $Y \rightarrow X$ CSR↑ | MSE↓ |
|---|---|---|---|---|---|---|---|---|---|
| 2k | NAIVE | - | 0.413 | - | 0.349 | - | 0.351 | - | 0.353 |
|  | MR-Egger | - | 1.993 | - | 2.067 | - | 0.852 | - | 0.981 |
|  | sisVIVE | 0.70 | 0.532 | 0.72 | 0.450 | 0.75 | 0.367 | 0.74 | 0.414 |
|  | IV-TETRAD | 0.63 | 0.822 | 0.35 | 0.792 | 0.68 | 0.651 | 0.29 | 0.716 |
|  | TSHT | 0.02 | 0.412 | 0.02 | 0.388 | 0.02 | 0.372 | 0.02 | 0.373 |
|  | PReBiM | **0.98** | **0.003** | **0.98** | **0.003** | **0.98** | **0.010** | **0.98** | **0.008** |
| 5k | NAIVE | - | 0.386 | - | 0.364 | - | 0.369 | - | 0.342 |
|  | MR-Egger | - | 2.145 | - | 2.570 | - | 0.943 | - | 0.952 |
|  | sisVIVE | 0.71 | 0.578 | 0.74 | 0.502 | 0.79 | 0.383 | 0.73 | 0.442 |
|  | IV-TETRAD | 0.66 | 0.811 | 0.33 | 0.899 | 0.68 | 0.714 | 0.29 | 0.596 |
|  | TSHT | 0.01 | 0.415 | 0.02 | 0.387 | 0.01 | 0.355 | 0.01 | 0.370 |
|  | PReBiM | **0.99** | **0.009** | **0.99** | **0.008** | **0.99** | **0.001** | **0.99** | **0.001** |
| 10k | NAIVE | - | 0.390 | - | 0.374 | - | 0.336 | - | 0.364 |
|  | MR-Egger | - | 2.015 | - | 2.902 | - | 1.051 | - | 0.994 |
|  | sisVIVE | 0.74 | 0.538 | 0.72 | 0.497 | 0.78 | 0.384 | 0.78 | 0.437 |
|  | IV-TETRAD | 0.65 | 0.812 | 0.35 | 0.877 | 0.74 | 0.630 | 0.25 | 0.792 |
|  | TSHT | 0.02 | 0.423 | 0.02 | 0.398 | 0.01 | 0.335 | 0.01 | 0.387 |
|  | PReBiM | **0.99** | **0.001** | **0.979** | **0.000** | **0.99** | **0.008** | **0.99** | **0.009** |

Note: The symbol "−" means that methods do not output this information. "↑" means a higher value is better, and vice versa.

