# OpenReview forum: "Identification and Estimation of the Bi-Directional MR with Some Invalid Instruments"
_NeurIPS.cc/2024/Conference — NeurIPS 2024 oral_

### Official Review · Reviewer_UQEn · 2024-07-12

**Soundness:** 3
**Presentation:** 3
**Contribution:** 3
**Rating:** 6
**Confidence:** 3

**Summary:**

The paper addresses the challenge of estimating causal effects in bi-directional Mendelian randomization (MR) studies using observational data, where invalid instruments and unmeasured confounding are common. It investigates theoretical conditions for identifying valid instrumental variable (IV) sets and proposes a cluster fusion-like algorithm to discover these IV sets and estimate causal effects accurately. Experimental results demonstrate the effectiveness of the method in handling bi-directional causal relationships, providing insights crucial for improving causal inference in complex systems.

**Strengths:**

The main contribution of the paper is presenting sufficient and necessary conditions for the identifiability of the bi-directional model, enabling both valid IV sets for each direction. They also propose a practical and effective cluster fusion-like algorithm for unbiased estimation based on the theorems and prove the correctness of the algorithm. The paper also validates the theoretical findings using extensive experiments on synthetic data along with comparisons to baseline methods. Overall, the paper is well written and easy to follow.

**Weaknesses:**

The paper has no major weaknesses. However, the setting is restricted with causal relations limited to being linear and assuming genetic variants are randomized, which limits the practical applicability of the proposed approach. Additionally, the experimental results provided are mainly synthetic in nature.

**Questions:**

I have few Questions/Suggestions for the Authors:

* In line 106, the authors mention, "Following Hausman [1983], we assume that $\beta_{X ->Y} \beta_{X->X} \neq 1$." It would be useful to discuss in the main paper why this assumption is necessary and what happens when it is violated.

* Similarly, regarding Assumption 3, the authors mention it as a very natural condition that one expects to hold for the unique identifiability of valid IVs. It would be useful to explain briefly in the main paper why this assumption is necessary for the identifiability of IVs.

* The authors in Section 5 claim that with dependence between genetic variants, main results may still be effective in identifying valid IV sets. Does this claim still hold when there is confounding among the genetic variants or between the genetic variants and some phenotype? Or does the dependence just mean direct causal effect here?.

* At the moment, the proposed solution is restricted to linear causal relationships. Can one apply the proposed method using some linearization technique for scenarios where causal relationships are not necessarily linear?

**Limitations:**

The authors clearly state all the assumptions. The paper could benefit from adding some more discussion on the necessity of these assumptions in the main paper. I don't think the paper has any potential negative societal impacts.

---

> ### Author Rebuttal · Authors · 2024-08-07
>
> We truly appreciate your insightful and encouraging comments. Please see below for our responses.
>
> >**W1. The setting is restricted with causal relations limited to being linear.**
>
> We’d like to mention that,
> - Identifying instrumental variables in bidirectional MR, both theoretically and practically, within the one-sample MR framework is a desirable but challenging research topic. We employed the **linearity assumption to entail the theoretical identifiability** of the bi-directional MR model, while the linearity model could enjoy some remarkable characteristics.
> - **Linear models have also been widely explored and used** in many practical situations [Pearl, 2009, Spirtes et al., 2000, Imbens and Rubin, 2015], often providing meaningful results [Kang et al., 2016, Windmeijer et al., 2021, Silva and Shimizu, 2017, Li and Ye, 2022]. Hence, we focus primarily on linear models, other than nonlinear ones. We leave nonlinear models to be our future works.
>
> >**W2. the experimental results are mainly synthetic in nature.**
>
> We performed experiments on two real-world datasets. One is derived from a study on causal relationships btw. obesity and Vitamin D Status [Vimaleswaran et al., 2013], while the other one from an empirical study on the impact of colonial history on the economic development of various regions [Acemoglu et al., 2001].
> - The first bi-directional dataset is produced based on the GWAS summary data from [Vimaleswaran et al., 2013] and a publicly available website. With obesity (X) and Vitamin D Status (Y), we selected 16 related SNPs as candidate IVs. They are fat mass and obesity-associated-rs9939609(FTO), Fas apoptotic inhibitory molecule 2-rs7138803(FAIM2), 7-dehydrocholesterol reductase-rs12785878(DHCR7), cytochrome P450 family 24 subfamily A member 1-rs6013897(CYP24A1), etc. We entail FTO and FAIM2 to be the valid IVs related to $X\to Y$ while DHCR7 and CYP24A1 are related to $Y\to X$. These results are in accordance with findings [Vimaleswaran et al., 2013].
> - The second one-directional dataset, the Colonial Origins dataset, consists of Institutions(X), and Economic Development(Y), with other 8 variables as candidate IVs [Acemoglu et al., 2001]. They are Latitude (lat_abst), European settlements in 1900 (euro1900), Log European settler mortality(logem4), etc. We find that our method selects euro1900 and logem4 as valid IVs, with the estimated causal effect 0.861, which are both consistent with results in [Acemoglu et al., 2001]. Will add the data details and results.
>
> >**Q1. In line 106, why $\beta_{X\to Y}\beta_{Y\to X}\neq1$ is necessary and what happens when violated.**
>
> We’d like to clarify if $\beta_{X \to Y}\beta_{Y\to X} = 1$, causal effects $\beta_{X\to Y}$ and $\beta_{Y\to X}$ are not identifiable, even given the valid IV. This condition serves as the fundamental identification criterion for Eq.(1). For more details, please see pages 402-407 of [Hausman, 1983]. Will explain it.
>
> >**Q2. Why Assumption 3 is necessary for the identifiability of IVs?**
>
> Note that according to Proposition 3, we obtain the IV set for one of the causal relationships, either $X\to Y$ or $Y\to X$. To achieve full identifiability of valid IVs in a bi-directional MR model, we need to further determine which causal direction the IV set is related to. Thus, to render the causal effects identifiable, we introduce Assumption 3 [Xue and Pan, 2020]. It employs the correlations between IVs and phenotypes to find the related direction for the IV set. We have included the necessity of assumptions in the revision.
>
> >**Q3-W3. genetic variants are randomized...dependence btw. genetic variants. Does this claim still hold when there is confounding among the genetic variants or between the genetic variants and some phenotype? Or does the dependence just mean direct causal effect here?**
>
> Thanks for your valuable comments. We conjecture that the claim still holds when there is confounding among the genetic variants or between the genetic variants and some phenotype. Following the example's proof in Appendix D, one can prove some simple examples of these cases (Due to space limitations, we do not provide detailed proofs here but will offer them in the revision).
> - In addition to the example proofs, we also performed empirical experiments to validate the first case. In Table 2 of the supplemented PDF, we see that when there’s confounding between genetic variants, our method remains effective with different sample sizes.
> - It's worth noting particularly that when there's confounding between a genetic variant pointing to X and X (not Y), our conjecture still holds; when the confounding is between the genetic variant pointing to X and Y, we would need this confounding factor to be observable to control the conditional independence between the genetic variant and Y. Will leave such general research into our future work.
>
> >**Q4. ...using some linearization technique for scenarios where causal relationships are not necessarily linear?**
>
> This is a very good point. When causal relationships are not necessarily linear, one can practically apply some linearization technique, mapping nonlinear causal relationships to possibly linear ones. In such a case with possibly linearity data, our conclusions might be still feasible. And we leave it as our future work, i.e., how to deal with not necessarily linear data as well as complex nonlinear data. Will add this discussion.
>
> **References**
>
> Judea Pearl. Causality: Models, Reasoning, and Inference. Cambridge University Press, New York, 2nd edition, 2009.
>
> Peter Spirtes, Clark Glymour, and Richard Scheines. Causation, Prediction, and Search. MIT press, 2000.
>
> Guido W Imbens and Donald B Rubin. Causal inference for statistics, social, and biomedical sciences: An introduction. Cambridge University Press, 2015.
>
> Daron Acemoglu, Simon Johnson, and James A. Robinson. The colonial origins of comparative development: An empirical investigation. American economic review, 2001.

---

> > ### Comment · Reviewer_UQEn · 2024-08-08
> > **Re.**
> >
> > Thanks for responding to my questions. The suggested changes by the authors, including additional experiments and clarifications, would be very beneficial for the paper. I will keep my decision and score for the paper.

---

### Official Review · Reviewer_St5u · 2024-07-12

**Soundness:** 4
**Presentation:** 4
**Contribution:** 4
**Rating:** 7
**Confidence:** 4

**Summary:**

The authors take up a _very useful_ topic, of trying to identify instruments in models where bidirectional adjacencies exist, at least for the Mendelian randomization application.

**Strengths:**

The topic of the paper is on point--this is something we need to know more about, as bidirectional edges obviously exist in real data.

This was an excellent paper, thanks. The discussion made sense to me from start to finish, and the experimental results were compelling. Thanks.

**Weaknesses:**

From _my_ perspective, there were no glaring weaknesses to this paper. Perhaps other reviewers have issues to mention.

The only possible weakness I saw was the strong reliance on the assumption of linearity, though in the discussion this was mentioned as an assumption that could possibly be relaxed in future work.

**Questions:**

No particular questions.

**Limitations:**

I did not see a discussion of societal impact.

---

> ### Author Rebuttal · Authors · 2024-08-06
>
> Thank you very much for your inspiring commendation. We would like to mention that:
>
> (i) Identifying instrumental variables in bidirectional MR, both theoretically and practically, within the one-sample MR framework is a desirable but challenging research topic. We employed the **linearity assumption to entail the theoretical identifiability** of the bi-directional MR model, while the linearity model could enjoy some remarkable characteristics.
>
> (ii) **Linear models have also been widely explored and used** in many practical situations [Pearl, 2009, Spirtes et al., 2000, Imbens and Rubin, 2015], often providing meaningful results [Kang et al., 2016, Windmeijer et al., 2021, Silva and Shimizu, 2017, Li and Ye, 2022]. Hence, we focus primarily on linear models, other than nonlinear ones.
>
> Furthermore, how to develop a framework to **handle nonlinear causal relationships** is a significant future direction.
>
>
>
>
> **References**
>
> Judea Pearl. Causality: Models, Reasoning, and Inference. Cambridge University Press, New York, 2nd edition, 2009.
>
> Peter Spirtes, Clark Glymour, and Richard Scheines. Causation, Prediction, and Search. MIT press, 2000.
>
> Guido W Imbens and Donald B Rubin. Causal inference for statistics, social, and biomedical sciences: An introduction. Cambridge University Press, 2015.

---

> > ### Comment · Reviewer_St5u · 2024-08-12
> > **Thanks.**
> >
> > Thanks for the rebuttal. I will stick to my original assessment.

---

### Official Review · Reviewer_XbcF · 2024-07-13

**Soundness:** 3
**Presentation:** 3
**Contribution:** 3
**Rating:** 6
**Confidence:** 4

**Summary:**

The paper addresses the problem of estimating causal effects in bi-directional Mendelian randomization (MR) models with some invalid instrumental variables (IVs) and unmeasured confounding. It proposes a framework for identifying valid IV sets under the assumption that the IV set consists of genetic variants that are independent of each other and that at least two of them are valid IVs.  The authors introduce a cluster fusion-like algorithm based on this framework and demonstrate its effectiveness through theoretical proofs and experimental results.

**Strengths:**

The authors establish both necessary and sufficient conditions for identifying bi-directional Mendelian randomization (MR) models, which builds upon previous work focusing on uni-directional MR.

The proposed cluster fusion-like algorithm is well-founded. The experimental results on synthetic datasets show the algorithm's efficacy in estimating causal effects. These results support the theoretical claims and suggest that the method performs well in practice.

**Weaknesses:**

While the paper discusses various assumptions (such as the independence of genetic variants and existence of two valid IVs), it would benefit from a more in-depth exploration of the limitations and potential pitfalls of these assumptions in real-world data. Addressing how violations of these assumptions impact the results could strengthen the paper.

The experiments are performed on synthetic datasets. While this is a good starting point, additional validation on real-world datasets would provide more robust evidence of the method’s practical utility.

**Questions:**

1) How does your method perform when the assumptions (e.g., independence of genetic variants) are violated in practice? Are there any robust techniques or adjustments to handle such cases?

2) Regarding the construction of the IV set, do you find that a larger IV set generally leads to more robust causal estimates, or does it introduce more complexity and potential for bias with invalid instruments?

3) Is the process of constructing the IV set dependent on the order in which instruments are considered? Specifically, does sequentially adding IVs versus a simultaneous assessment of all potential IVs impact the validity and effectiveness of the identified set?

4) Can the proposed algorithm handle large-scale datasets efficiently? What are the computational complexities and potential bottlenecks?

**Limitations:**

See weaknesses.

---

> ### Author Rebuttal · Authors · 2024-08-07
>
> Thank you for your insightful comments and suggestions. We have addressed the comments related to the empirical experiment and three assumptions in real-world data. Please see our responses below.
>
> >**W1.** ...it would benefit from a more in-depth exploration of the limitations...how violations of these assumptions impact the results could strengthen the paper.
>
> - **Assumption 1** can be easily justified in practice, since it allows the number of valid IVs to equal 2 for the bi-directional model which is much milder than existing methods. Please see examples in real-world experiments.
> - Honestly, it is hard to test **Assumptions 2 or 3** directly in real life, since usually we cannot obtain the ground truths of causal effects between any two variables, including latent confounders. It should be noted that Assumption 2 is satisfied mostly in reality as the set of conditions that meet this assumption occupies a very small portion of the entire space, making it very demanding to violate. If Assumption 3 is violated, we may fail to determine the causal direction for the identified IV set.
>
> >**W2.** real-world datasets?
>
> We additionally evaluated our method on two real-world datasets. One is derived from a study on the bi-directional causal relationships between obesity and Vitamin D Status [1], while the other one from an empirical study on the impact of colonial history on the economic development of various regions [2].
>
> - The first bi-directional dataset is based on GWAS summary data from [1] and a publicly available website. For obesity (X) and Vitamin D status (Y), we selected 16 related SNPs as candidate IVs, including FTO, FAIM2, DHCR7, and CYP24A1. FTO and FAIM2 are valid IVs for \(X \to Y\), while DHCR7 and CYP24A1 are valid IVs for \(Y \to X\), with causal effects of -1.15 and -0.05, respectively. These results align with findings in [1].
>
> - The second one-directional dataset, the Colonial Origins dataset, consists of Institutions(X), and Economic Development(Y), with other 8 variables as candidate IVs [2]. They are Latitude, European settlements in 1900 (euro1900), Log European settler mortality(logem4), etc. We find that our method selects euro1900 and logem4 as valid IVs, with the estimated causal effect 0.861, which are both consistent with results in [2].  Will add the data details and results.
>
> We will add the data description details and results in the revisions.
>
> [1] Vimaleswaran K S., et al. Causal relationship between obesity and vitamin D status: bi-directional Mendelian randomization analysis of multiple cohorts. PLoS Med, 2013.
>
> [2] Acemoglu D, et al. The colonial origins of comparative development: An empirical investigation. Am Econ Rev, 2001.
>
> >**Q1.** How does your method perform when the assumptions (e.g., independence of genetic variants) are violated in practice?
>
> **A1:** First, we conducted experiments with dependent genetic variants on both bi-directional and one-directional MR data, as illustrated in Appendix H.1-H.2. Table 3 shows that our method performs superiorly across various sample sizes and scenarios, even with correlated genetic variants.
>
> Second, we performed empirical experiments with confounding among genetic variants. As shown in Table 2 of the supplementary PDF, our method remains effective across different sample sizes.
>
> These results highlight its capability to accurately identify effective IVs and provide consistent causal effect estimates from observational data, regardless of assumption violations. This implies no need for adjustment.
>
> >**Q2.** a larger IV set or does it introduce more complexity and potential for bias with invalid instruments?
>
> **A2:** Thank you for your question.
> - A larger IV set can sometimes offer a broader range of IVs to better capture variations in the treatment variables, potentially enhancing the robustness of the estimates. However, the size of the IV set alone does not guarantee its validity for causal inference. Even with a large IV set, it might not effectively address unobserved confounders, which can introduce significant biases into the estimation results [3]. When constructing IV sets, it is crucial to ensure that the selected IVs are strongly correlated with the treatment variable while remaining conditionally independent of the outcome variable. If these conditions are not satisfied, robust causal estimates may remain elusive, regardless of the IV set size.
>
> - Moreover, a large IV set might introduce more complexity for our method (see the complexity of our method in the next answer). Therefore, introducing additional parameters, such as W in Algorithm 1, to control the set size can help manage complexity and reduce potential bias.
>
> [3] Zawadzki R S., et al. Frameworks for estimating causal effects in observational settings: comparing confounder adjustment and instrumental variables. BMC Med Res Methodol, 2023.
>
> >**Q3.** constructing the IV set dependent on the order in which instruments are considered?
>
> **A3:** We would like to clarify that our algorithm does not depend on the order of the candidate IVs. As demonstrated in Lines 3-4 and 9-11 of Algorithm 4 in Appendix E, we evaluate simultaneously all subsets of IVs and compute their corresponding correlations, ultimately selecting the subset with the minimum correlation. It ensures the robustness of the algorithm.
>
> >**Q4.** large-scale datasets efficiently?  computational complexities?
>
> **A4:** In summary, the computational complexity of our PReBiM algorithm is:
>
>  $\sum_{k=0}^{t} (2\binom{g-kW}{2} + \frac{(2g-(2k+1)W-2)(W-1)}{2}) + 2W(t-1)$,
>
> where g is the number of IVs, W is the maximum length of the IV set, and t is the number of loops.
>
> For validation, we performed synthetic experiments on S(10,10,30) and S(15,15,40), with 30 and 40 IVs, respectively. Results are shown in Table 1 of the supplementary PDF. We observe that the overall performance of all methods decreases with larger-scale IVs, but our method still outperforms the baselines.

---

> > ### Comment · Reviewer_XbcF · 2024-08-12
> > **Thank you for the responses.**
> >
> > Thank you for your detailed responses. I appreciate the clarification and will maintain my positive score.

---

### Official Review · Reviewer_Qa6w · 2024-07-14

**Soundness:** 3
**Presentation:** 4
**Contribution:** 3
**Rating:** 7
**Confidence:** 4

**Summary:**

This paper studies the identifiability problem of the bi-directional Mendelian randomization (MR) model, where $X$ and $Y$ are a pair of phenotypes of interest and causes of each other, and $\textbf{G}$ is the set of measured genetic variants, which may include invalid instrumental variables (IVs). Under some assumptions, the paper has identified and proved correct the sufficient and necessary conditions for identifying valid IV sets from $\textbf{G}$  based on observational data, without requiring prior knowledge about which candidate IVs in $\textbf{G}$ are valid or invalid. Supported by the theoretical result, an algorithm is proposed for finding the valid IV sets from the set of measured genetic variants  $\textbf{G}$ using observation data and estimating the bi-directional causal effects using the found valid IVs. Experiments are conducted with synthetic data to show the effectiveness of the proposed algorithm.

**Strengths:**

1. The paper addresses a challenging and practical problem.
2. The work is comprehensive, with both theoretical results and corresponding algorithm presented.
3. The paper is very well written in general.

**Weaknesses:**

1. The experimental evaluation is done with synthetic data only. Although the presented experiments with synthetic data are comprehensive and the identification conditions have been theoretically proved, as the theoretical result relies on several assumptions, it would be necessary to conduct some case studies with real world data to evaluate how the method works in practice (where domain knowledge or literature can be used to justify the correctness of the found IV sets)

2. It would be very helpful if the assumptions and their feasibility (and consequences/limitations) in practice can be illustrated and justified with real world examples.

**Questions:**

1. Line 106: Does the assumption regarding the multiplication of the two effects have any practical meaning/implication?
2. Could you explain what "cluster fusion" means exactly in the paper and why the proposed algorithm is said to be "cluster fusion-like"?
3. Section 6.2 - how the one-directional data used in this section generated?
4. The work is based on the assumed structure in Figure 2 (plus some invalid IVs as illustrated in the other figures) , but in practice there would be more complicated situations than those, e.g. the vertical pleiotropy effect in biology where the IVs (genetic variants) are associated with another phenotype (or biological pathway) and this in turn causes the two phenotypes of interest ($X$ and $Y$).

**Limitations:**

Some limitations of the paper have been discussed briefly, but as mentioned above, the consequence and limitations due to the assumptions should be discussed a bit more.

---

> ### Author Rebuttal · Authors · 2024-08-07
>
> We appreciate your time dedicated to reviewing our paper and your thoughtful and encouraging comments. Below, please see our responses. We hope they can resolve your concerns. Note that we also summarize the main concerns of all reviewers.  Please refer to the general response if interested.
>
> >**W1. it's necessary to conduct some case studies with real world data.**
>
> We additionally evaluated our method on two real-world datasets. One is derived from a study on the bi-directional causal relationships between obesity and Vitamin D Status [1], while the other one from an empirical study on the impact of colonial history on the economic development of various regions [2].
> - This first bi-directional dataset is produced based on the GWAS summary data from [1] and a publicly available website. With obesity (X) and Vitamin D Status (Y), we selected 16 related SNPs as candidate IVs, they are fat mass and obesity-associated-rs9939609(FTO), Fas apoptotic inhibitory molecule 2-rs7138803(FAIM2), 7-dehydrocholesterol reductase-rs12785878(DHCR7), cytochrome P450 family 24 subfamily A member 1-rs6013897(CYP24A1), etc. We entail FTO and FAIM2 to be the valid IVs related to $X\to Y$ while DHCR7 and CYP24A1 are valid IVs related to $Y\to X$, with causal effects -1.15 and -0.05, respectively. These results are in accordance with findings [1].
> - The second one-directional dataset, the Colonial Origins dataset, consists of Institutions(X), and Economic Development(Y), with other 8 variables as candidate IVs [2]. They are Latitude (lat_abst), European settlements in 1900 (euro1900), Log European settler mortality(logem4), etc. We find that our method selects euro1900 and logem4 as valid IVs, with the estimated causal effect 0.861, which are both consistent with results in [2]. Will add the data details and results.
>
> >**W2. the assumptions and their feasibility (and consequences/limitations) in practice.**
>
> - **Assumption 1** can be easily justified in practice, since it allows the number of valid IVs to equal 2 for the bi-directional model which is much milder than existing methods. Please see examples in real-world experiments.
> - Honestly, it is hard to test **Assumptions 2 or 3** directly in real life, since usually we cannot obtain the ground truths of causal effects between any two variables, including latent confounders. It should be noted that Assumption 2 is satisfied mostly in reality as the set of conditions that meet this assumption occupies a very small portion of the entire space, making it very demanding to violate. If Assumption 3 is violated, we may fail to determine the causal direction for the identified IV set. Will add them.
>
> >**Q1: Line 106: Does the assumption regarding the multiplication of the two effects have any practical meaning/implication?**
>
> We would like to clarify that if $\beta_{X \to Y} \beta_{Y \to X} = 1$, the causal effects $\beta_{X \to Y}$ and $\beta_{Y \to X}$ are not identifiable, even given the valid IV. In fact, this condition serves as the fundamental identification criterion for Eq.(1). For more detailed information, please refer to pages 402-407 of [Hausman, 1983]. We will include this discussion in the revision.
>
> >**Q2: Could you explain what "cluster fusion" means exactly in the paper and why the proposed algorithm is said to be "cluster fusion-like"?**
>
> A cluster is considered a valid IV set. The term "fusion-like" suggests a specific process for identifying and merging these clusters. We will add the explanation.
>
> >**Q3: Section 6.2 - how the one-directional data used in this section generated?**
>
> The one-directional data in Section 6.2 is simply generated by setting $\beta_{Y \to X} = 0$ in Eq.(11), shown below. Will emphasize it in the revision.
>
> $$U=\mathbf{G}^\intercal\gamma_U+\varepsilon_1,X=\mathbf{G}^\intercal\gamma_X+U\gamma_{X,U}+\varepsilon_2,$$
> $$Y=X\beta_{X\to Y}+\mathbf{G}^\intercal\gamma_Y+U\gamma_{Y,U}+\varepsilon_3,$$
> $$G_{ij}\sim Binomial(2,maf_j),maf_j\sim\mathcal{U}(0.1,0.5). \tag{11}$$
>
> >**Q4: ...complicated situations, e.g. the vertical pleiotropy effect in biology where the IVs (genetic variants) are associated with another phenotype (or biological pathway) and this in turn causes the two phenotypes of interest (X and Y).**
>
> Thanks for the insightful idea. When it comes to the complicated structure with the vertical pleiotropy effect from IV (denote another phenotype as T), we could find that such an IV still satisfies Assumption A2 [Exclusion Restriction] once given T. So we could upgrade Definition 1 of Pseudo-Residual conditional T, where $\omega_{\mathbb{G}}$ is obtained by Two-Stage Least Squares (TSLS) estimator but also needs to be conditional on T. We will add it with an example in Section 5 Discussion and regard it as our future work. Thanks again.
>
> **References**
>
> [1] Vimaleswaran K S, Berry D J, Lu C, et al. Causal relationship between obesity and vitamin D status: bi-directional Mendelian randomization analysis of multiple cohorts[J]. PLoS medicine, 2013, 10(2): e1001383.
>
> [2] Acemoglu D, Johnson S, Robinson J A. The colonial origins of comparative development: An empirical investigation[J]. American economic review, 2001, 91(5): 1369-1401.

---

> > ### Comment · Reviewer_Qa6w · 2024-08-12
> > **Thanks for your responses**
> >
> > Thanks the authors for your detailed responses. The extra experiments and discussions will be very helpful. I am happy to keep my positive rating.

---

### Author Rebuttal · Authors · 2024-08-07

We thank all reviewers for their constructive suggestions and **overall positive comments**, especially for the acknowledgment of our writing quality, comprehensive theoretical analysis, and empirical experimental performance.

We have taken carefully the reviewers' feedback into account and responded to each question with detailed explanations and additional experimental results. Please see below the summarized main concerns.

# Experiments
Following the suggestions from reviewers, we provide additional results on two **real-world datasets** to further validate the effectiveness of our method and enhance our paper. One is derived from a study on the bi-directional causal relationships between obesity and Vitamin D Status [1], while the other one from an empirical study on the impact of colonial history on the economic development of various regions [2]. Experimental results on both datasets revealed that our method could find valid IVs as well as obtain causal effects, which are consistent with those findings from existing literature.

Moreover, we performed additional **synthetic experiments** to validate the efficacy of our method, with **results shown in the supplemented PDF**.

# Assumptions
Following the suggestions from reviewers, we provide in-depth discussion and exploration of the necessity of assumptions, to strengthen the paper.

- Compared with existing methods that constrain the number of valid IVs to be larger than 2, **Assumption 1** of our method allows the number of valid IVs to equal 2 for the bi-directional model, which is much milder. If it is violated, our method as well as other mentioned methods would fail to identify a valid IV set theoretically.

- Note that **Assumption 2** is satisfied mostly in reality as the set of conditions that meet this assumption occupies a very small portion of the entire space, making it very demanding to violate.

- To derive the full identifiability of valid IVs in our bi-directional MR model, i.e., determining which causal direction the IV set is related to, we further introduce **Assumption 3**. If it is violated, we may fail to determine the causal direction for the identified IV set.

- **Linearity assumption**. Identifying instrumental variables in bidirectional MR, both theoretically and practically, within the one-sample MR framework is a desirable but challenging research topic. We employed the linearity assumption to entail the theoretical identifiability of the bi-directional MR model, while the linearity model could enjoy some remarkable characteristics.

We sincerely thank the reviewers and the AC for their time and thoughtful feedback on our paper. We hope that our responses have effectively addressed all the questions and concerns.

**References**

[1]Vimaleswaran K S, Berry D J, Lu C, et al. Causal relationship between obesity and vitamin D status: bi-directional Mendelian randomization analysis of multiple cohorts[J]. PLoS medicine, 2013, 10(2): e1001383.

[2] Acemoglu D, Johnson S, Robinson J A. The colonial origins of comparative development: An empirical investigation[J]. American economic review, 2001, 91(5): 1369-1401.

---

### Author Response · Authors · 2024-08-13
**Thank all reviewers again for greatly improving our paper!**

Dear all reviewers,

We sincerely appreciate all your positive comments! We are grateful for your valuable and inspiring suggestions, which are of great help in improving our paper!

Best wishes,

The authors

---

### Decision · Program_Chairs · 2024-09-25

**Decision:**

Accept (oral)

**Comment:**

The paper addresses the challenging problem of causal effect estimation from observational data in Mendelian randomization (MR). It studies the identification problem in bi-directional MR and presents complete theoretical results with both necessary and sufficient conditions. Additionally, the paper introduces an algorithm and reports experimental results on both synthetic and real-world datasets.

All reviewers agree that the work tackles a challenging and practical problem. The theoretical results are sound and complete, the algorithm is well-founded, and the evaluations are comprehensive. However, there was some concern about the initial lack of experiments on real-world datasets. In response, the authors added two new experiments involving real-world data in the discussion phase. Reviewers were also interested in understanding the impact of the strong assumptions on the feasibility of the algorithm. The authors provided explanations for these assumptions, which were satisfactory to the reviewers.